# Accelerating Rescaled Gradient Descent: Fast Optimization of Smooth Functions

**Ashia C. Wilson**
Microsoft Research
ashia.wilson@microsoft.com

**Lester Mackey**
Microsoft Research
lmackey@microsoft.com

**Andre Wibisono**
Georgia Tech
wibisono@gatech.edu

## Abstract

We present a family of algorithms, called *descent algorithms*, for optimizing convex and non-convex functions. We also introduce a new first-order algorithm, called *rescaled gradient descent* (RGD), and show that RGD achieves a faster convergence rate than gradient descent over the class of *strongly smooth* functions – a natural generalization of the standard smoothness assumption on the objective function. When the objective function is convex, we present two frameworks for accelerating descent algorithms, one in the style of Nesterov and the other in the style of Monteiro and Svaiter, using a single Lyapunov function. Rescaled gradient descent can be accelerated under the same strong smoothness assumption using both frameworks. We provide several examples of strongly smooth loss functions in machine learning and numerical experiments that verify our theoretical findings. We also present several extensions of our novel Lyapunov framework including deriving optimal universal higher-order tensor methods and extending our framework to the coordinate descent setting.

## 1 Introduction

We consider the optimization problem

$$\min_{x \in \mathcal{X}} f(x) \tag{1}$$

where $f : \mathcal{X} \to \mathbb{R}$ is a continuously differentiable function, on a finite-dimensional real vector space $\mathcal{X}$ with inner product norm $\|v\| := \sqrt{\langle v, Bv \rangle}$ and a dual norm $\|s\|_* := \sqrt{\langle s, B^{-1}s \rangle}$ for $s$ in the dual space $\mathcal{X}^*$. Here, $B : \mathcal{X} \to \mathcal{X}^*$ is a positive definite self-adjoint operator. We assume the minimum of $f$ is attainable and let $x^*$ represent a point in $\arg\min_{x \in \mathcal{X}} f(x)$.

We study the performance of a family of discrete-time algorithms parameterized by $\delta > 0$ and an integer scalar $1 < p \le \infty$, called *$\delta$-descent algorithms of order $p$*. These algorithms meet a progress condition that allows us to derive fast non-asymptotic convergence rate upper bounds, parameterized by $p$, for both nonconvex and convex instances of (1). For example, descent algorithms of order $1 < p < \infty$ satisfy the upper bound $f(x_k) - f(x^*) = O(1/(\delta k)^{p-1})$ for convex functions.

Using this framework we introduce a new method for smooth optimization called *rescaled gradient descent* (RGD),

$$x_{k+1} = x_k - \eta^{\frac{1}{p-1}} \frac{B^{-1}\nabla f(x_k)}{\|\nabla f(x_k)\|_*^{\frac{p-2}{p-1}}}, \qquad \eta > 0, p > 1.$$

We show that if (1) is sufficiently smooth, rescaled gradient descent is a $\delta$-descent algorithm of order $p$, and subsequently converges quickly to solutions of (1). RGD can be viewed as a natural generalization of gradient descent ($p = 2$) and normalized gradient descent ($p = \infty$), whose non-asymptotic behavior for quasi-convex functions has been well-studied ([11]).

When $f$ is convex, we present two frameworks for obtaining algorithms with faster convergence rate upper bounds. The first, pioneered in Nesterov [22, 23, 24, 25], shows how to wrap a $\delta$-descent method of order $1 < p < \infty$ in two sequences to obtain a method that satisfies $f(x_k) - f(x^*) = O(1/(\delta k)^p)$. The second, introduced by [18], shows how to wrap a $\delta$-descent method of order $1 < p < \infty$ in the same set of sequences and add a line search step to obtain a method that satisfies $f(x_k) - f(x^*) = O(1/(\delta k)^{\frac{3p-2}{2}})$. We provide a general description of both frameworks and show how they can be applied to RGD and other descent methods of order $p$.

Our motivation also comes from a burgeoning literature (e.g., [27, 28, 30, 33, 13, 35, 4, 8, 32, 29, 31, 17]) that harnesses the connection between dynamical systems and optimization algorithms to develop new analyses and optimization methods. Rescaled gradient descent is obtained by discretizing an ODE called *rescaled gradient flow* introduced by [34]. We compare RGD and accelerated RGD to the work of Zhang et al. [36], who introduce accelerated dynamics and apply Runge-Kutta integrators to discretize them. They show that Runge-Kutta integrators converge quickly when the function is sufficiently smooth and when the order of the integrator is sufficiently large. We provide a better convergence rate upper bound for accelerated RGD under a very similar smoothness assumption. We also compare our work to Maddison et al. [17], who introduces conformal Hamiltonian dynamics and show that if the objective function is sufficiently smooth, algorithms obtained by discretizing these dynamics converge at a linear rate. We show (accelerated) RGD also achieves a fast linear rate under similar smoothness conditions.

The remainder of this paper is organized as follows. Section 2 introduces $\delta$-descent algorithms and Section 2.1 describes several examples of descent algorithms that are popular in optimization. Section 2.2 introduces RGD and Section 3 presents two frameworks for accelerating $\delta$-descent methods and applies both to RGD. Section 5 describes several examples of strongly smooth objective functions as well as experiments to verify our findings. Finally, Section 6 discusses simple extensions of our framework, including deriving and analyzing *optimal universal tensor methods* for objective functions that have Hölder-continuous higher-order gradients and extending our entire framework and results to the coordinate setting.

## 2 Descent Algorithms

The focus of this section is a family of algorithms called $\delta$-*descent algorithms of order $p$*.

**Definition 1** *An algorithm $x_{k+1} = \mathcal{A}(x_k)$ is a $\delta$-**descent algorithm of order** $p$ for $1 < p \leq \infty$ if for some constant $0 < \delta < \infty$ it satisfies*

$$\frac{f(x_{k+1}) - f(x_k)}{\delta} \leq -\|\nabla f(x_k)\|_*^{\frac{p}{p-1}} \quad \text{for all } k \geq 0, \text{ or} \tag{2a}$$

$$\frac{f(x_{k+1}) - f(x_k)}{\delta} \leq -\|\nabla f(x_{k+1})\|_*^{\frac{p}{p-1}} \quad \text{for all } k \geq 0. \tag{2b}$$

For $\delta$-descent algorithms of order $p$, it is possible to obtain non-asymptotic convergence guarantees for non-convex, convex and gradient dominated functions. Recall, a function is $\mu$-*gradient dominated of order $p \in (1, \infty]$* if

$$\frac{p-1}{p}\|\nabla f(x)\|_*^{\frac{p}{p-1}} \geq \mu^{\frac{1}{p-1}}(f(x) - f(x^*)), \quad \forall x \in \mathcal{X}. \tag{3}$$

When $p = 2$, (3) is the Polyak-Łojasiewicz condition introduced concurrently by Polyak [27] and Łojasiewicz [16]. For the following three theorems, we use the shorthand $E_0 := f(x_0) - f(x^*)$ and assume $f$ is differentiable.

**Theorem 1** *Any $\delta$-descent algorithm of order $p$ satisfies*

$$\min_{0 \leq s \leq k} \|\nabla f(x_s)\|_* \leq (E_0/(\delta k))^{\frac{p-1}{p}}. \tag{4}$$

**Theorem 2** *If $f$ is convex with $R = \sup_{x:f(x)\leq f(x_0)} \|x - x^*\| < \infty$, and $c_p := \frac{(1-1/p)^p}{p-1}$, then any $\delta$-descent algorithm of order $p$ satisfies*

$$f(x_k) - f(x^*) \leq \begin{cases} 2\left(\frac{1}{E_0^{1/p}} + \frac{1}{R\gamma c_p^{1/p}p}(\delta k)^{\frac{p-1}{p}}\right)^{-p} = O(1/(1 + \frac{1}{R\gamma p}(\delta k)^{\frac{p-1}{p}})^p), & p < \infty \\ 2E_0 \exp(-\delta k/(R\gamma)), & p = \infty. \end{cases} \tag{5}$$

*where $\gamma = 1$ when (2a) is satisfied and $\gamma = (1 + \frac{1}{Rp}(E_0/c_p)^{\frac{1}{p}}\delta^{\frac{p-1}{p}})^{p-1}$ when (2b) is satisfied.*

**Theorem 3** *If $f$ is $\mu$-gradient dominated of order $p$, then any $\delta$-descent algorithm of order $p$ satisfies*

$$f(x_k) - f(x^*) \leq E_0 \exp\left(-\tfrac{p}{p-1}\mu^{\frac{1}{p-1}}\delta k\right). \tag{6}$$

The proof of Theorems 1 to 3 are all based on simple energy arguments and can be found in Appendix B. Bounds of the form (4) are common in the non-convex optimization literature and have previously been established for gradient descent ($p = 2$ see e.g. [26, Thm1]) and higher-order tensor methods (see e.g.[6]). Theorem 1 provides a more general description of algorithms that satisfy this kind of bound.

Typically, algorithms satisfy the progress condition (2) for specific smoothness classes of functions. For example, gradient descent with step-size $0 < \eta \leq 1/L$ is a $\delta$-descent method of order $p = 2$ with $\delta = \eta/2$ when $\|\nabla^2 f\| \leq L$. Throughout, we denote $\|B\| = \max_{\|h\|\leq 1}\|Bh\|_*$, for any $B : \mathcal{X} \to \mathcal{X}^*$. We list several other examples.

## 2.1 Examples of descent algorithms

Theorems 1, 2 and 3 provide a seamless way to derive standard upper bounds for many algorithms in optimization.

**Example 1** *The* universal higher-order tensor method,

$$x_{k+1} = \arg\min_{x\in\mathcal{X}}\left\{f_{p-1}(x; x_k) + \frac{1}{\tilde{p}\eta}\|x - x_k\|^{\tilde{p}}\right\}, \tag{7}$$

*where $f_{p-1}(y; x) = \sum_{i=0}^{p-1}\frac{1}{i!}\nabla^i f(x)(y-x)^i$ is the $(p-1)$-st order Taylor approximation of $f$ centered at $x$ and $\tilde{p} = p - 1 + \nu$ for $\nu \in (0,1]$, has been studied by several works [3, 34, 21]. When $f$ is convex and has Hölder-smooth $(p-1)$-st order gradients, namely $\|\nabla^{p-1}f(x) - \nabla^{p-1}f(y)\| \leq L\|x-y\|^\nu$, (7) with step size $0 < \eta \leq \frac{\sqrt{3}(p-2)!}{2L}$, is a $\delta$-descent algorithm of order $\tilde{p}$ with $\delta = \eta^{\frac{1}{\tilde{p}-1}}/2^{\frac{2\tilde{p}-3}{\tilde{p}-1}}$.*

**Example 2** The natural proximal method,

$$x_{k+1} = \arg\min_{x\in\mathcal{X}}\left\{f(x) + \frac{1}{p\eta}\|x - x_k\|_{x_k}^p\right\}, \tag{8}$$

*where $\|v\|_x = \sqrt{\langle v, \nabla^2 h(x)v\rangle}$ was introduced in the setting $h(x) = \frac{1}{2}\|x\|_2^2$ by [19]. For any $\eta, m > 0$ and $mB \preceq \nabla^2 h$, the proximal method is a $\delta$-descent algorithm of order $p$ with $\delta = m^{\frac{p}{p-1}}\eta^{\frac{1}{p-1}}/p$.*

**Example 3** Natural gradient descent,

$$x_{k+1} = x_k - \eta\nabla^2 h(x_k)^{-1}\nabla f(x_k) = \arg\min_{x\in\mathcal{X}}\left\{\langle\nabla f(x_k), x\rangle + \frac{1}{2\eta}\|x - x_k\|_{x_k}^2\right\}, \tag{9}$$

*where $\|v\|_x = \sqrt{\langle v, \nabla^2 h(x)v\rangle}$ was introduced by [2]. Suppose $\|\nabla^2 f\| \leq L$ and $mB \preceq \nabla^2 h \preceq MB$ for some $m, L, M > 0$. Then natural gradient descent with step size $0 < \eta \leq \frac{m^2}{ML}$ is a $\delta$-descent algorithm of order $p = 2$ with $\delta = \frac{\eta}{2M}$.*

**Example 4** Mirror descent,

$$x_{k+1} = \arg\min_{x\in\mathcal{X}}\left\{\langle\nabla f(x_k), x\rangle + \frac{1}{\eta}D_h(x, x_k)\right\}, \tag{10}$$

*where $D_h(x, y) = h(x) - h(y) - \langle\nabla h(y), x - y\rangle$ is the Bregman divergence was introduced by [20]. Suppose $\|\nabla^2 f\| \leq L$ and $mB \preceq \nabla^2 h \preceq MB$ for some $m, L, M > 0$. Then mirror descent with step size $0 < \eta \leq \frac{m^2}{ML}$ is a $\delta$-descent algorithm of order $p = 2$ with $\delta = \frac{\eta}{2M}$.*

**Example 5** The proximal Bregman method,

$$x_{k+1} = \arg\min_{x\in\mathcal{X}}\left\{f(x) + \frac{1}{\eta}D_h(x, x_k)\right\}, \tag{11}$$

*was introduced by [7]). When $mB \preceq \nabla^2 h \preceq MB$ the proximal Bregman method with step-size $\eta > 0$ is a $\delta$-descent algorithm of order $p = 2$ with $\delta = \frac{m\eta}{2M^2}$.*

Details for these examples are contained in Appendix B.2.

## 2.2 Rescaled gradient descent

We end this section by discussing the function class for which rescaled gradient descent (RGD),

$$x_{k+1} = x_k - \eta^{\frac{1}{p-1}} \frac{B^{-1}\nabla f(x_k)}{\|\nabla f(x_k)\|_*^{\frac{p-2}{p-1}}} = \arg\min_{x\in\mathcal{X}} \left\{ \langle \nabla f(x_k), x\rangle + \frac{1}{p\eta}\|x - x_k\|^p \right\}, \quad (12)$$

is a $\delta$-descent method of order $p$.

**Definition 2** *A function $f$ is **strongly smooth** of order $p$ for some integer $p > 1$, if there exist constants $0 < L_1, \ldots, L_p < \infty$ such that for $m = 1, \ldots, p - 1$ and for all $x \in \mathbb{R}^d$:*

$$|\nabla^m f(x)(B^{-1}\nabla f(x))^m| \le L_m \|\nabla f(x)\|_*^{m+\frac{p-m}{p-1}} \quad (13)$$

*and moreover for $m = p$, $f$ satisfies the condition $|\nabla^p f(x)(v)^p| \le L_p \|v\|^p, \forall v \in \mathcal{X}$.*

Here, $\nabla^m f(x)(h)^m = \sum_{i_1,\ldots,i_m=1}^d \partial_{x_{i_1}\ldots x_{i_m}} f(x) \prod_{j=1}^m h_{i_j}$ where $\partial_{x_i} f$ is the partial derivative of $f$ with respect to $x_i$. We can always take $L_1 = 1$. When $p = 2$, (13) is the usual Lipschitz condition on the gradient of $f$, but otherwise (13) is stronger. In particular, if $f$ is strongly smooth of order $p$, then the minimizer $x^*$ has order at least $p - 1$, i.e., the higher gradients vanish: $\nabla^m f(x^*) = 0$ for $m = 1, \ldots, p - 1$, whereas this is not implied under mere smoothness. An example of a strongly smooth function of order $p$ is the $p$-th power of the $\ell_2$-norm $f(x) = \|x\|_2^p$ with $B = I$, or the $\ell_p$-norm $f(x) = \|x\|_p^p$. We discuss other families of strongly smooth functions in Section 5. Finally, it is worth mentioning that for most of our results, the absolute value on the left hand side of (13) is unnecessary. We now present the main result regarding the performance of RGD on functions that satisfy (13):

**Theorem 4** *Suppose $f$ is strongly smooth of order $p > 1$ with constants $0 < L_1, \ldots, L_p < \infty$. Then rescaled gradient descent with step-size*

$$0 < \eta^{\frac{1}{p-1}} \le \min\left\{1, \frac{1}{\left(2\sum_{m=2}^p \frac{L_m}{m!}\right)}\right\} \quad (14)$$

*satisfies the descent condition (2a) with $\delta = \eta^{\frac{1}{p-1}}/2$.*

The proof of Theorem 4 is in Appendix B.3. A corollary to Theorems 1-4 is the following theorem.

**Theorem 5** *RGD with a step size that satisfies (14) achieves convergence rate guarantee (4) when $f$ is differentiable and strongly smooth of order p, (5) when $f$ is convex function and strongly smooth of order p, and (6) when $f$ is $\mu$-uniformly convex and strongly smooth of order p, where $\delta^{p-1} = \eta/2^{p-1}$.*

Our results show rescaled gradient descent can minimize the canonical $p$-strongly smooth and uniformly convex function $f(x) = \frac{1}{p}\|x\|^p$ at an exponential rate; in contrast, gradient descent can only minimize it at a polynomial rate, even in one dimension. We provide the proof of Proposition 6 in Appendix B.4.

**Proposition 6** *Let $f: \mathbb{R} \to \mathbb{R}$ be $f(x) = \frac{1}{p}|x|^p$ for $p > 2$, with minimizer $x^* = 0$ and $f(x^*) = 0$. For any step size $0 < \eta^{\frac{1}{p-1}} < 1$ and initial position $x_0 \in \mathbb{R}$, rescaled gradient descent of order $p$ minimizes $f$ at an exponential rate: $f(x_k) = (1 - \eta^{\frac{1}{p-1}})^{pk} f(x_0)$. On the other hand, for any $\eta^{\frac{1}{p-1}} > 0$ and $|x_0| < (2\eta^{\frac{1}{p-1}})^{-\frac{1}{p-2}}$, gradient descent minimizes $f$ at a polynomial rate: $f(x_k) = \Omega((\eta^{\frac{1}{p-1}}k)^{-\frac{p}{p-2}})$.*

We now demonstrate how all the aforementioned examples of $\delta$-descent methods can be accelerated.

## 3 Accelerating Descent Algorithms

We present two frameworks for accelerating descent algorithms based on the dynamical systems perspective introduced by Wibisono et al. [34] and Wilson et al. [35] and apply them to RGD. The backbone of both frameworks is the Lyapunov function

$$E_k = A_k(f(x_k) - f(x^*)) + D_h(x^*, z_k),$$

and two sequences (15) and (16). The connection between continuous time dynamical systems and these two sequences and Lyapunov function is described in [35]. We present a high-level description of both techniques in the main text and leave details of our analysis to Appendix C.

### 3.1 Nesterov acceleration of descent algorithms

In the context of convex optimization, the technique of "acceleration" has its origins in Nesterov [22] and refined in Nesterov [23]. In these works, Nesterov showed how to combine gradient descent with two sequences to obtain an algorithm with an optimal convergence rate. There have been many works since (as well as some frameworks, including [15, 1, 14, 35]) describing how to accelerate various other algorithms to obtain methods with superior convergence rates.

Wilson et al. [35], for example, show the following two discretizing schemes,

$$x_k = \delta\tau_k z_k + (1 - \delta\tau_k)y_k \tag{15a}$$

$$z_{k+1} = \arg\min_{z \in \mathcal{X}} \left\{ \alpha_k \langle \nabla f(x_k), z \rangle + \tfrac{1}{\delta} D_h(z, z_k) \right\} \tag{15b}$$

where $y_{k+1}$ satisfies the $\delta^{\frac{p}{p-1}}$-descent condition $f(y_{k+1}) - f(x_k) \le -\delta^{\frac{p}{p-1}} \|\nabla f(x_k)\|_*^{\frac{p}{p-1}}$; and

$$x_k = \delta\tau_k z_k + (1 - \delta\tau_k)y_k \tag{16a}$$

$$z_{k+1} = \arg\min_z \left\{ \alpha_k \langle \nabla f(y_{k+1}), z \rangle + \tfrac{1}{\delta} D_h(z, z_k) \right\}, \tag{16b}$$

where the update for $y_{k+1}$ satisfies the ($\delta^{\frac{p}{p-1}}$-descent) condition $f(y_{k+1}) - f(x_k) \le \langle \nabla f(y_{k+1}), y_{k+1} - x_k \rangle \le -\delta^{\frac{p}{p-1}} \|\nabla f(y_{k+1})\|_*^{\frac{p}{p-1}}$, constitute an "accelerated method". Their results can be summarized in the following theorem.

**Theorem 7** *Assume for all $x, y \in \mathcal{X}$, the function $h$ satisfies the local uniform convexity condition $D_h(x, y) \ge \frac{1}{p}\|x - y\|^p$. Then sequences (15) and (16) with parameter choices $\alpha_k = (\delta/p)^{p-1} k^{(p-1)}$ (where $k^{(p)} := k(k+1) \cdots (k + p - 1)$ is the rising factorial) and $\tau_k = \frac{p}{\delta(p+k)} = \Theta(\frac{p}{\delta k})$ satisfy,*

$$f(y_k) - f(x^*) \le \frac{p^p D_h(x^*, z_0)}{(\delta k)^p} = O\left(1/(\delta k)^p\right). \tag{17}$$

Proof details are contained in Appendix C.1. Wilson et al. [35] call these new methods *accelerated descent methods* due to the fact that Theorem 2 guarantees implementing just the $y_{k+1}$ sequence (where we set $x_k = y_k$) satisfies $f(y_k) - f(x^*) \le O(1/(\tilde{\delta}k)^{p-1})$, where $\tilde{\delta}^{p-1} = \delta^p$. The computational cost of adding sequences (15a) and (15b) (or (16a) and (16b)) to the descent method is at most an additional gradient evaluation.

**Remark 1 (Restarting for accelerated linear convergence)** *If, in addition, $f$ is $\mu$-gradient dominated of order $p$, then algorithms (15) and (16) combined with a scheme for restarting the algorithm has a convergence rate upper bound $f(y_k) - f(x^*) = O(\exp(-\mu^{\frac{1}{p}}\delta k))$. We can consider this algorithm an accelerated method given the original descent method satisfies $f(y_k) - f(x^*) = O(\exp(-\mu^{\frac{1}{p-1}}\tilde{\delta}k))$ under the same condition, where $\tilde{\delta}^{p-1} = \delta^p$. See Appendix C.2 for details.*

To summarize, it is sufficient to establish conditions under which an algorithm is a $\delta$-descent algorithm of order $p$ in order to (1) obtain a convergence rate and (2) accelerate the algorithm (in most cases).

**Accelerated rescaled gradient descent (Nesterov-style)**   Using (15) we accelerate RGD.

---

**Algorithm 1** Nesterov-style accelerated rescaled gradient descent.

---

**Require:** $f$ satisfies (13) and $h$ satisfies $D_h(x, y) \ge \frac{1}{p}\|x - y\|^p$
1: Set $x_0 = z_0$, $A_k = (\delta/p)^p k^{(p)}$, $\alpha_k = \frac{A_{k+1} - A_k}{\delta}$, $\tau_k = \frac{\alpha_k}{A_{k+1}}$, and $\delta^{\frac{p}{p-1}} = \eta^{\frac{1}{p-1}}/2$.
2: **for** $k = 1, \ldots, K$ **do**
3: $\quad x_k = \delta\tau_k z_k + (1 - \delta\tau_k)y_k$
4: $\quad z_{k+1} = \arg\min_{z \in \mathcal{X}} \left\{ \alpha_k \langle \nabla f(x_k), z \rangle + \tfrac{1}{\delta} D_h(z, z_k) \right\}$
5: $\quad y_{k+1} = x_k - \eta^{\frac{1}{p-1}} B^{-1} \nabla f(x_k) / \|\nabla f(x_k)\|_*^{\frac{p-2}{p-1}}$
6: **return** $y_K$.

---

We summarize the performance of Algorithm 1 in the following Corollary to Theorems 4 and 7:

**Theorem 8** *Suppose $f$ is convex and strongly smooth of order $1 < p < \infty$ with constants $0 < L_1, \ldots, L_p < \infty$. Also suppose $\eta$ satisfies (14). Then Algorithm 1 satisfies the convergence rate upper bound (17).*

## 3.2 Monteiro-Svaiter acceleration of descent algorithms

Recently, Monteiro and Svaiter [18] have introduced an alternative framework for accelerating descent methods, which is similar to Nesterov's scheme but includes a line search step. This framework was further generalized by several more recent concurrent works [9, 12, 5] who demonstrate that higher-order tensor method (7) with the addition of a line search step obtains a convergence rate upper bound $f(y_k) - f(x^*) = O(1/k^{\frac{3p-2}{2}})$. When $p = 2$, this rate matches that of the Nesterov-style acceleration framework, but for $p > 2$ it is better. In this section, we present a novel, generalized version of the Monteiro-Svaiter accleration framework. In particular, *we use a simple Lyapunov analysis to generalize the framework* and show that many other descent methods of order $p$ can be accelerated in it, including the proximal method (8), RGD (12) and universal tensor methods.

**Theorem 9** *Suppose $h$ is satisfies the condition $B \preceq \nabla^2 h$. Consider sequence (15) where in addition, we add a line search step which ensures the inequalities*

$$a \leq \frac{\lambda_{k+1}}{\delta^{\frac{3p-2}{2}}} \|y_{k+1} - x_k\|^{p-2} \leq b, \quad 0 < a < b \quad and \tag{18a}$$

$$\|y_{k+1} - x_k + \lambda_{k+1} \nabla f(y_{k+1})\| \leq \tfrac{1}{2} \|y_{k+1} - x_k\| \tag{18b}$$

*hold for the pair $(\lambda_{k+1}, y_{k+1})$, where $\lambda_{k+1} = \delta^2 \alpha_k^2 / A_{k+1}$. Then the composite sequence satisfies:*

$$f(y_k) - f(x^*) \leq \frac{p^{\frac{3p-2}{2}} D_h(x^*, x_0)^{\frac{p}{2}}}{(\delta k)^{\frac{3p-2}{2}}} = O\left(1/(\delta k)^{\frac{3p-2}{2}}\right). \tag{19}$$

The proof of Theorem 9 is in Appendix C.3. All the aforementioned concurrent works have demonstrated that the higher-order gradient method ($\nu = 1$) with the addition line search step satisfies (18). We show the same is true of the proximal method (8), rescaled gradient descent (12) and universal higher-order tensor methods. See Appendix C.5 for details. We conjecture that all methods that satisfy conditions (18a) and (18b) are descent methods of order $p$ with an additional line search step.

**Remark 2 (Restarting for improved accelerated linear rate)** *If, in addition, $f$ is $\mu$-gradient dominated of order $p$, then (18) combined with a scheme for restarting the algorithm satisfies the convergence rate upper bound $f(y_k) - f(x^*) = O(\exp(-\mu^{\frac{2}{3p-2}} \delta k))$. See Appendix C.2 for details.*

## 3.3 Accelerating rescaled gradient descent (Monteiro-Svaiter-style)

Monteiro-Svaiter accelerated rescaled gradient descent is the following algorithm.

---

**Algorithm 2** Monteiro-Svaiter-style accelerated rescaled gradient descent.

---

**Require:** $f$ is *strongly smooth of order* $1 < p < \infty$ and $h$ satisfies $B \preceq \nabla^2 h$.
1: Set $x_0 = z_0 = 0$, $A_0 = 0$, $\delta^{\frac{3p-2}{2}} = \eta, \eta^{\frac{1}{p-1}} \leq \min\{\frac{2}{5p}, 1/(2\sum_{m=2}^p \frac{L_m}{m!})\}$
2: **for** $k = 1, \ldots, K$ **do**
3: Choose $\lambda_{k+1}$ (e.g. by line search) such that $\frac{3}{4} \leq \frac{\lambda_{k+1}\|y_{k+1} - x_k\|^{p-2}}{\eta} \leq \frac{5}{4}$, where

$$y_{k+1} = x_k - \eta^{\frac{1}{p-1}} \frac{\nabla f(x_k)}{\|\nabla f(x_k)\|_*^{\frac{p-2}{p-1}}},$$

and $\alpha_k = \frac{\lambda_{k+1} + \sqrt{\lambda_{k+1} + 4A_k \lambda_{k+1}}}{2\delta}$, $A_{k+1} = \delta\alpha_k + A_k$, $\tau_k = \frac{\alpha_k}{A_{k+1}}$ (so that $\lambda_{k+1} = \frac{\delta^2 \alpha_k^2}{A_{k+1}}$) and

$$x_k = \delta\tau_k z_k + (1 - \delta\tau_k)y_k.$$

4: Update $z_{k+1} = \arg\min_{z \in \mathcal{X}} \left\{\alpha_k \langle \nabla f(y_{k+1}), z\rangle + \frac{1}{\delta} D_h(z, z_k)\right\}$
5: **return** $y_K$.

---

We summarize results on performance of Algorithm 2 in the following corollary to Theorem 9:

**Theorem 10** *Assume $f$ is convex and strongly smooth of order $1 < p < \infty$ with constants $0 < L_1, \ldots, L_p < \infty$. Then Algorithm 2 satisfies the convergence rate upper bound* (19).

## 4 Related Work

Our acceleration framework is similar in spirit to a number of acceleration frameworks in the literature (e.g., Allen Zhu and Orecchia [1], Lessard et al. [14], Lin et al. [15], Diakonikolas and Orecchia [8]) but applies more generally to descent methods of order $p > 2$. In particular, the present framework builds off of the framework proposed by Wilson et al. [35], but it (1) makes the connection to descent methods more explicit and (2) incorporates a generalization and Lyapunov analysis of the Monteiro-Svaiter acceleration framework. These manifold generalizations crucially allow us to propose RGD and accelerated RGD, which has superior theoretical and empirical performance to several existing methods on strongly smooth functions.

## 5 Examples and Numerical Experiments

We compare our result to several recent works that have shown that for some function classes, more intuitive first-order algorithms outperform gradient descent. In particular, both Zhang et al. [36] and Maddison et al. [17] obtain first-order algorithms by applying integration techniques to second-order ODEs. When the objective function is sufficiently smooth, both show their algorithm outperforms (accelerated) gradient descent. We show that Algorithms 1 and 2 achieves fast performance in theory and in practice on similar objectives.

**Runge-Kutta** Zhang et al. [36] show that if one applies an $s$-th order Runge-Kutta integrator to a family of second-order dynamics, then the resulting algorithm[1] achieves a convergence rate $f(x_k) - f(x^*) = O(1/k^{\frac{ps}{s-1}})$[2] provided the function meet the following two conditions: (1) $f$ satisfies the *gradient lower bound* of order $p \geq 2$, which means for all $m = 1, \ldots, p-1$,

$$f(x) - f(x^*) \geq \tfrac{1}{C_m} \|\nabla^m f(x)\|^{\frac{p}{p-m}} \quad \forall \, x \in \mathbb{R}^n \tag{21}$$

for some constants $0 < C_1, \ldots, C_{p-1} < \infty$; and (2) for $s \geq p$ and $M > 0$, $f$ is $(s+2)$-times differentiable and $\|\nabla^{(i)} f(x)\| \leq M$ for $i = p, p+1, \ldots, s+2$. One can show that if $f$ is strongly smooth of order $p$, then $f$ satisfies the gradient lower bound of order $p$. The details of this result is in D. While are unable to prove that condition (21) is equivalent to strong smoothness, we have yet to find an example of a function that satisfies (21) and is not strongly smooth.

**Hamiltonian Descent** Maddison et al. [17] show explicit integration techniques applied to conformal Hamiltonian dynamics converge at a fast linear rate for a function class larger than gradient descent. The method entails finding a kinetic energy map that upper bounds the dual of the function. All examples for which we can compute such a map given by [17] are uniformly convex and gradient dominated functions; therefore, simply rescaling the gradient for these examples ensures a linear rate.

### 5.1 Examples

We provide several examples of strongly smooth functions in machine learning (see Appendix D.2 for details).

**Example 6** *The $\ell_p$ loss function*

$$f(x) = \tfrac{1}{p} \|Ax - b\|_p^p, \tag{22}$$

*shown by Zhang et al. [36] to satisfy* (21) *of order $p$, is strongly smooth of order $p$.*

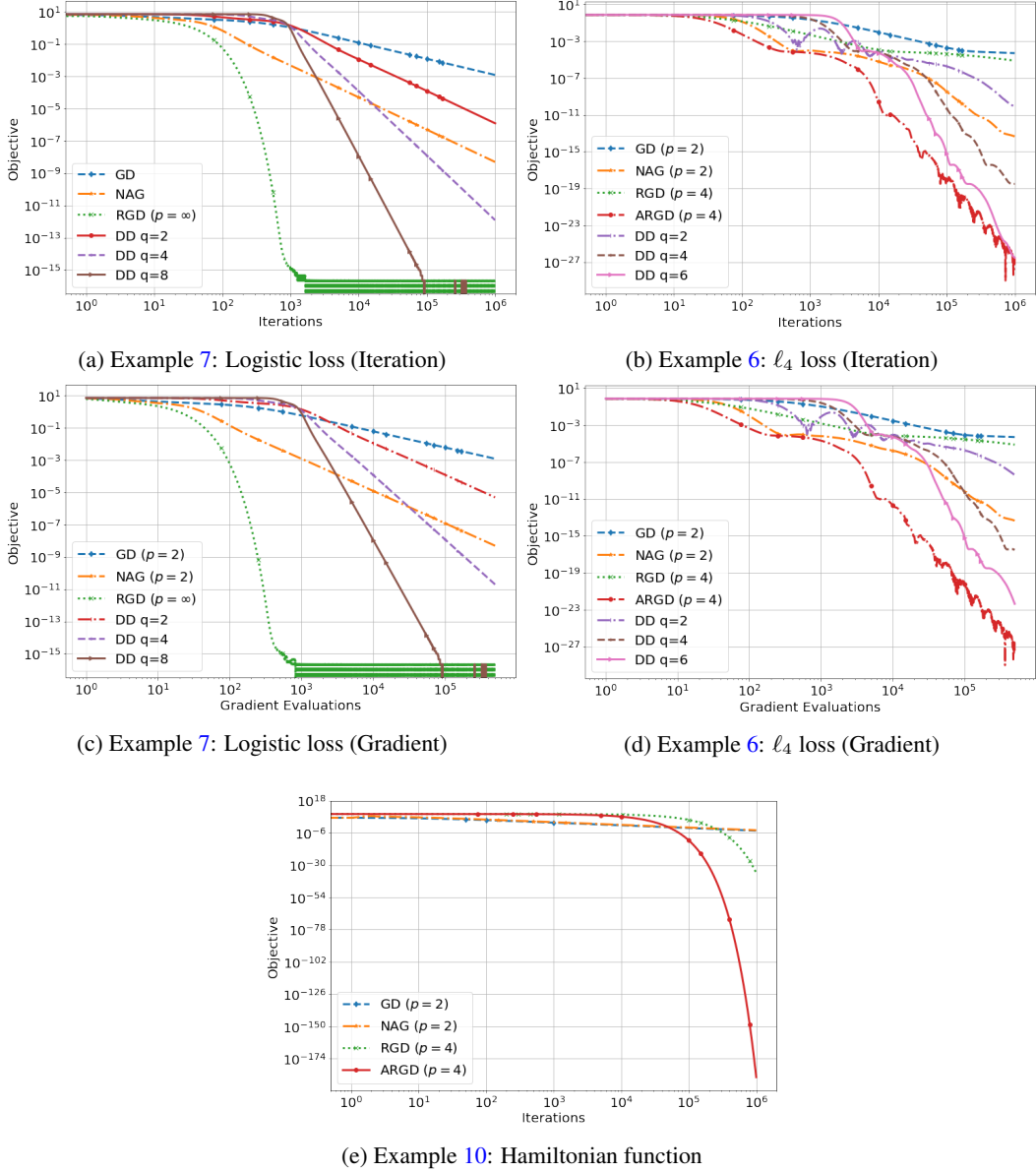

(a) Example 7: Logistic loss (Iteration)

(b) Example 6: $\ell_4$ loss (Iteration)

(c) Example 7: Logistic loss (Gradient)

(d) Example 6: $\ell_4$ loss (Gradient)

(e) Example 10: Hamiltonian function

Figure 1: Experimental results comparing RGD and accelerated RGD (ARGD) to gradient descent (GD), Nesterov accelerated GD (NAG) and Runge-Kutta (DD). The plots for Runge-Kutta use an $s = 2$ integrator which requires two gradient evaluations per iteration. Where relevant, we plot both iterations (Figs. 1a and 1b) and gradient evaluations (Figs. 1c and 1d).

**Example 7** *The logistic loss*

$$f(x) = \log(1 + e^{-yw^\top x}), \tag{23}$$

*shown by Zhang et al. [36] to satisfy (21) of order $p = \infty$, is strongly smooth of order $p = \infty$.*

**Example 8** *The GLM loss,*

$$f(x) = \tfrac{1}{2}(y - \phi(x^\top w))^2 \quad for \quad \phi(r) = 1/(1 + e^{-r}), \quad y \in \{0,1\}, \quad and \quad w \in \mathbb{R}^d, \tag{24}$$

*studied by Hazan et al. [11] is strongly smooth of order $p = 3$.*

**Example 9** *The $\ell_2$ loss to the $p$-th power*

$$f(x) = \tfrac{1}{p}\|Ax - b\|_2^p, \tag{25}$$

*for which Hamiltonian descent [17] obtains a linear rate, is strongly smooth and gradient dominated of order $p$.*

**Example 10** *The loss function,*

$$f(x) = (x^{(1)} + x^{(2)})^4 + \tfrac{1}{16}(x^{(1)} - x^{(2)})^4, \tag{26}$$

*for which Hamiltonian descent [17] obtains a linear rate, is strongly smooth and gradient dominated of order $p = 4$.*

## 5.2 Experiments

In this section, we perform a series of numerical experiments to compare the performance of ARGD (Algorithm 1) with gradient descent (GD), Nesterov accelerated GD (NAG), and the state-of-the-art Runge-Kutta algorithms of Zhang et al. [36] (DD) on the logistic loss $f(x) = \sum_{i=1}^{10} \log(1 + e^{-w_i^\top x y_i})$, the $\ell_4$ loss $f(x) = \frac{1}{4}\|Ax - b\|_4^4$, and the Hamiltonian descent loss (Example 10). For the logistic and $\ell_4$ losses, we use the same code, plots, and experimental methodology of Zhang et al. [36] (including data and step-size choice), adding to it (A)RGD. Specifically, for Fig. 1a-Fig. 1d, the entries of $W \in \mathbb{R}^{10 \times 10}$ and $A \in \mathbb{R}^{10 \times 10}$ are i.i.d. standard Gaussian, and the first five entries of $y$ (and $b$) are valued 0 while the rest are 1. Fig. 1e shows the performance of A(RGD), GD, and NAG on the Hamiltonian objective studied by [17]; for Fig. 1e, the largest step-size was chosen subject to the algorithm not diverging. For each experiment, a simple implementation of (A)RGD significantly outperforms the Runge-Kutta algorithm (DD), GD and NAG. The code for these experiments can be found here: `https://github.com/aswilson07/ARGD.git`.

# 6 Additional Results and Discussion

This paper establishes broad conditions under which an algorithm will converge and its performance can be accelerated by adding momentum. We use these conditions to introduce (accelerated) rescaled gradient descent for strongly smooth functions, and showed it outperforms several recent first-order methods that have been introduced for optimizing smooth functions in machine learning.

There are (at least) two simple extensions of our framework. First, an analogous framework can be established for **(accelerated) $\delta$-coordinate descent methods** of order $p$. As an application, we introduce (accelerated) rescaled coordinate descent for functions that are strongly smooth along each coordinate direction of the gradient. We provide details in Appendix E.1. Second, with our generalization of the Monteiro-Svaiter framework, we derive **optimal univeral tensor methods** for functions whose $(p - 1)$-st gradients are $\nu$-Hölder-smooth which achieve the upper bound $f(y_k) - f(x^*) = O(1/k^{\frac{3\tilde{p}-2}{2}})$ where $\tilde{p} = p - 1 + \nu$. The matching lower bound for this class of functions was recently established by [10]. We present this result in Appendix E.3.

There are several possible directions for future work. We know that certain simple operations preserve convexity (e.g., addition), but what operations preserve strong smoothness? Understanding this could allow us to construct more complex examples of strongly smooth functions. Our results reveal an interesting hierarchy of smoothness assumptions which lead to methods that converge quickly; exploring this more is of significant interest. Finally, extending our analysis to the stochastic or manifold setting, studying the use of variance reduction techniques, and introducing other $\delta$-decent algorithms of order $p$ are all interesting directions for future work.

**Acknowledgments**

We would like to thank Jingzhao Zhang for providing us access to his code.

## Footnotes

[1] which requires at least $s$ gradient evaluations per iteration

[2] this matches the rate of Algorithm 1 in the limit $s \to \infty$, where $s$ is the order of the integrator.

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
