[Supplementary Material]


# Accelerating Rescaled Gradient Descent: Fast Minimization of Smooth Functions

Ashia C. Wilson    Lester Mackey    Andre Wibisono

## A    Descent Flows

The derivation and analysis of descent algorithms is inspired by *descent flows*. In this section we introduce and analyzed these family of dynamics.

**Definition 3** *A dynamics is a* **descent flow of order** *$p$ if is satisfies*

$$\frac{d}{dt}f(X_t) \leq -\|\nabla f(X_t)\|_*^{\frac{p}{p-1}},\tag{27}$$

*for some $1 < p \leq \infty$ and for all $0 \leq t \leq \infty$.*

For dynamics that satisfy (27), we obtain non-asymptotic convergence guarantees for non-convex, convex and gradient-dominated functions. We summarize our main results for descent curves of order $p$ in the following three theorems.

**Theorem 11** *Suppose a dynamical system satisfies* (27) *for some $1 < p \leq \infty$ and $f$ is differentiable. Then the system satisfies*

$$\min_{0 \leq s \leq t} \|\nabla f(X_s)\|_* = O\left(1/t^{\frac{p-1}{p}}\right).\tag{28}$$

**Theorem 12** *Suppose a dynamical system satisfies* (27) *for some $1 < p \leq \infty$ and $f$ is differentiable and convex with $R = \sup_{x:f(x) \leq f(x_0)} \|x - x^*\| < \infty$. Then the system satisfies*

$$f(X_t) - f(x^*) = \begin{cases} O\left(1/\left(1 + \frac{1}{Rp}t^{\frac{p-1}{p}}\right)^p\right) & \text{if } p < \infty \\ O\left(e^{-t/R}\right) & \text{if } p = \infty \end{cases}.\tag{29}$$

**Theorem 13** *Suppose a dynamical system satisfies* (27) *for some $1 < p \leq \infty$ and $f$ is differentiable and $\mu$-gradient dominated of order $p$. Then the system satisfies*

$$f(X_t) - f(x^*) = O\left(e^{-\frac{p}{p-1}\mu^{\frac{1}{p-1}}t}\right).\tag{30}$$

The proof of these results follows the same structure as the descent algorithms, with both relying on simple energy arguments.

### A.1    Proofs

To show (28), we begin with the energy function $\mathcal{E}_t = f(X_t) - f(x^*)$. A straightforward calculation shows

$$\frac{d}{dt}\mathcal{E}_t = \frac{d}{dt}f(X_t) \overset{(27)}{\leq} -\|\nabla f(X_t)\|_*^{\frac{p}{p-1}}.$$

Integrating and rearranging gives the bound

$$t \min_{0 \leq s \leq t} \|\nabla f(X_s)\|^{\frac{p}{p-1}} \leq \int_0^t -\|\nabla f(X_t)\|_*^{\frac{p}{p-1}} dt \leq \mathcal{E}_0 - \mathcal{E}_t.$$

from which we can conclude (28).

Next, fix any $a > 0$, and define the positive increasing function $w_a(t) = (1 + t/(ap))^p$ which satisfies $\frac{d}{dt} \log w_a(t) = \frac{1}{aw_a(t)^{1/p}}$ and the constant $c_p = \frac{(1-1/p)^p}{p-1}$. When $p = \infty$, each formal expression written in terms of $p$ in this proof should be interpreted as the limit of that expression as $p \to \infty$. For example, if $p = \infty$, $w_a(t) = \lim_{q \to \infty}(1 + t/(aq))^q = e^{t/a}$ and $c_p = \lim_{q \to \infty} \frac{(1-1/q)^q}{q-1} = 0$.

To establish (29), we show the energy function
$$\mathcal{E}_t = w_a(t)(f(X_t) - f(x^*)) \tag{31}$$
grows at most linearly for any dynamical system that satisfies (27). To this end, observe that
$$\begin{aligned}
\frac{d}{dt}\mathcal{E}_t &= w_a'(t)(f(X_t) - f(x^*)) + w_a(t)\frac{d}{dt}f(X_t) \\
&\leq w_a'(t)\langle \nabla f(X_t), x^* - X_t \rangle + w_a(t)\frac{d}{dt}f(X_t) \\
&\overset{(27)}{\leq} w_a'(t)\langle \nabla f(X_t), x^* - X_t \rangle - w_a(t)\|\nabla f(X_t)\|_*^{\frac{p}{p-1}} \\
&= w_a(t)(\frac{d}{dt}\log w_a(t)\langle \nabla f(X_t), x^* - X_t \rangle - \|\nabla f(X_t)\|_*^{\frac{p}{p-1}}) \\
&\leq w_a(t)c_p\|\frac{d}{dt}\log w_a(t)(X_t - x^*)\|^p \\
&= c_p\|X_t - x^*\|^p/a^p \leq c_p\frac{R^p}{a^p}.
\end{aligned}$$
The first inequality uses the convexity of $f$ and the second inequality uses (27). The third inequality uses the Fenchel-Young inequality
$$-\|s\|_*^{\frac{p}{p-1}} + \langle s, u \rangle \leq c_p\|u\|^p \tag{32}$$
for $s = \nabla f(X_t)$ and $u = \frac{d}{dt}\log w_a(t)(x^* - X_t)$. The last step uses the fact that $\|X_t - x^*\| \leq R = \sup_{x:f(x) \leq f(X_0)}\|x - x^*\|$ since condition (27) implies the dynamical system is a descent method. Moreover, $R$ is finite, since the sublevel sets of $f$ are bounded. Integrating allows us to obtain the statement $\mathcal{E}_t - \mathcal{E}_0 \leq c_p\frac{R^p}{a^p}t$, and subsequently, the upper bound
$$f(X_t) - f(x^*) \leq \frac{f(X_0)-f(x^*)}{(1+t/(ap))^p} + c_p\frac{R^p}{a^p}\frac{t}{(1+t/(ap))^p}.$$

Since $a > 0$ was arbitrary, we may choose $a = R\frac{(c_pt)^{1/p}}{(f(X_0)-f(x^*))^{1/p}}$ to obtain the bound
$$f(X_t) - f(x^*) \leq \frac{2(f(X_0)-f(x^*))}{\left(1+\frac{(f(X_0)-f(x^*))^{1/p}}{Rc_p^{1/p}p}t^{\frac{p-1}{p}}\right)^p} = O(1/(1 + \frac{1}{Rp}t^{\frac{p-1}{p}})^p)$$

as desired.

The last bound (30) uses the energy function $\mathcal{E}_t = f(X_t) - f(x^*)$ to establish
$$\frac{d}{dt}\mathcal{E}_t = \frac{d}{dt}f(X_t) \overset{(27)}{\leq} -\|\nabla f(X_t)\|_*^{\frac{p}{p-1}} \leq \frac{p}{p-1}\mu^{\frac{1}{p-1}}\mathcal{E}_t.$$
where the last inequality follows from the gradient dominated condition. We use the intuition from the bounds established for descent dynamics to derive analogous results for descent algorithms.

# B   Descent Algorithms

We present proofs of results Section 2.

## B.1   Proof of Theorems 1-3

We begin with detailed proofs of Theorems 1-3.

### B.1.1   Proof of Theorem 1

By rearranging and summing (2), we obtain
$$\delta k \min_{j-k \leq s \leq j}\|\nabla f(x_s)\|_*^{\frac{p}{p-1}} \leq \sum_{s=j-k}^{j}\|\nabla f(x_s)\|_*^{\frac{p}{p-1}}\delta \leq f(x_0) - f(x_k) \leq f(x_0)$$
where $j = k$ if the bound (2a) holds and $j = k+1$ if the bound (2b) holds. Rearranging the inequality yields the result in Theorem 1.

### B.1.2 Proof of Theorem 2

Fix any $a > 0$, and define the positive increasing function $w_a(t) = (1 + t/(ap))^p$, which satisfies $\frac{d}{dt} \log w_a(t) = \frac{1}{a w_a(t)^{1/p}}$, and the constant $c_p = \frac{(1-1/p)^p}{p-1}$. When $p = \infty$, each formal expression written in terms of $p$ in this proof should be interpreted as the limit of that expression as $p \to \infty$. For example, if $p = \infty$, $w_a(t) = \lim_{q\to\infty}(1 + t/(aq))^q = e^{t/a}$ and $c_\infty = \lim_{q\to\infty}\frac{(1-1/q)^q}{q-1} = 0$. For the proof of Theorem 2 under the condition (2a), we introduce the energy function

$$E_k = w_a(\delta k)(f(x_k) - f(x^*)),$$

noting that, by the convexity of $w$ on $t \geq 0$,

$$\frac{w_a(\delta(k+1)) - w_a(\delta k)}{\delta} \leq \frac{1}{a}(1 + \frac{\delta(k+1)}{ap})^{p-1} = \frac{1}{a}w_a(\delta(k+1))^{(p-1)/p}.$$

and hence

$$\frac{w_a(\delta(k+1)) - w_a(\delta k)}{\delta w_a(\delta(k+1))} \leq \frac{1}{a w_a(\delta(k+1))^{1/p}}. \tag{33}$$

When (2a) holds, we have

$$
\begin{aligned}
\frac{E_{k+1} - E_k}{\delta} &= \frac{w_a(\delta(k+1)) - w_a(\delta k)}{\delta}(f(x_k) - f(x^*)) + w_a(\delta(k+1))\frac{f(x_{k+1}) - f(x_k)}{\delta} \\
&\leq \frac{w_a(\delta(k+1)) - w_a(\delta k)}{\delta}\langle\nabla f(x_k), x_k - x^*\rangle + w_a(\delta(k+1))\frac{f(x_{k+1}) - f(x_k)}{\delta} \\
&\overset{(2a)}{\leq} \frac{w_a(\delta(k+1)) - w_a(\delta k)}{\delta}\langle\nabla f(x_k), x_k - x^*\rangle - w_a(\delta(k+1))\|\nabla f(x_k)\|_*^{\frac{p}{p-1}} \\
&= w_a(\delta(k+1))\left(\frac{w_a(\delta(k+1)) - w_a(\delta k)}{\delta w_a(\delta(k+1))}\langle\nabla f(x_k), x_k - x^*\rangle - \|\nabla f(x_k)\|_*^{\frac{p}{p-1}}\right) \\
&\leq w_a(\delta(k+1))\left(\frac{1}{a w_a(\delta(k+1))^{1/p}}\langle\nabla f(x_k), x_k - x^*\rangle - \|\nabla f(x_k)\|_*^{\frac{p}{p-1}}\right) \\
&\leq w_a(\delta(k+1))c_p\|\frac{1}{a w_a(\delta(k+1))^{1/p}}(x_k - x^*)\|^p \\
&= c_p\|x_k - x^*\|^p/a^p \leq c_p R^p/a^p.
\end{aligned}
$$

The first inequality uses convexity of $f$, and the second uses (2a). The third inequality is an application of (33). The fourth inequality uses the Fenchel-Young inequality $-\|s\|^{\frac{p}{p-1}} + \langle s, u\rangle \leq -\frac{p-1}{p}\|s\|^{\frac{p}{p-1}} + \langle s, u\rangle \leq \frac{1}{p}\|u\|^p$ with $s = \nabla f(x_k)$ and $u = \frac{1}{a w_a(\delta(k+1))^{1/p}}(x_k - x^*)$. Both descent conditions (2) imply $\|x_k - x^*\| \leq R$, yielding the final inequality. Therefore, we have shown that for all $k \geq 0$, $E_{k+1} - E_k \leq c_p\delta R^p/a^p$. This implies $E_k \leq E_0 + c_p\delta k R^p/a^p$. Therefore

$$f(x_k) - f(x^*) \leq \frac{f(x_0) - f(x^*)}{(1 + \delta k/(ap))^p} + c_p\frac{R^p}{a^p}\frac{\delta k}{(1 + \delta k/(ap))^p}.$$

Since $a > 0$ was arbitrary, we may choose $a = R\frac{(c_p\delta k)^{1/p}}{(f(x_0) - f(x^*))^{1/p}}$ to obtain the bound

$$f(x_k) - f(x^*) \leq \frac{2(f(x_0) - f(x^*))}{\left(1 + \frac{(f(x_0) - f(x^*))^{1/p}}{R c_p^{1/p} p}(\delta k)^{\frac{p-1}{p}}\right)^p} = O(1/(1 + \frac{1}{Rp}(\delta k)^{\frac{p-1}{p}})^p)$$

as desired.

If, on the other hand (2b) holds, identical reasoning yields

$$
\begin{aligned}
\frac{E_{k+1} - E_k}{\delta} &= \frac{w_a(\delta(k+1)) - w_a(\delta k)}{\delta}(f(x_{k+1}) - f(x^*)) + w_a(\delta k)\frac{f(x_{k+1}) - f(x_k)}{\delta} \\
&\leq \frac{w_a(\delta(k+1)) - w_a(\delta k)}{\delta}\langle\nabla f(x_{k+1}), x_{k+1} - x^*\rangle + w_a(\delta k)\frac{f(x_{k+1}) - f(x_k)}{\delta} \\
&\overset{(2b)}{\leq} \frac{w_a(\delta(k+1)) - w_a(\delta k)}{\delta}\langle\nabla f(x_{k+1}), x_{k+1} - x^*\rangle - w_a(\delta k)\|\nabla f(x_{k+1})\|_*^{\frac{p}{p-1}} \\
&= w_a(\delta k)\left(\frac{w_a(\delta(k+1)) - w_a(\delta k)}{\delta w_a(\delta k)}\langle\nabla f(x_{k+1}), x_{k+1} - x^*\rangle - \|\nabla f(x_{k+1})\|_*^{\frac{p}{p-1}}\right) \\
&\leq w_a(\delta k)\left(\frac{w_a(\delta(k+1))}{a w_a(\delta k) w_a(\delta(k+1))^{1/p}}\langle\nabla f(x_{k+1}), x_{k+1} - x^*\rangle - \|\nabla f(x_{k+1})\|_*^{\frac{p}{p-1}}\right) \\
&\leq w_a(\delta k)c_p\|\frac{w_a(\delta(k+1))}{a w_a(\delta k) w_a(\delta(k+1))^{1/p}}(x_{k+1} - x^*)\|^p \\
&= \left(\frac{w_a(\delta(k+1))}{w_a(\delta k)}\right)^{p-1}c_p\frac{R^p}{a^p}.
\end{aligned}
$$

Now, since $w_a(\delta(k+1)) \le w_a(\delta k) w_a(\delta)$, we have shown that for all $k \ge 0$, $E_{k+1} - E_k \le w_a(\delta)^{p-1} c_p \frac{R^p}{a^p} \delta$. This implies $E_k \le E_0 + w_a(\delta)^{p-1} c_p \frac{R^p}{a^p} \delta k$. Hence, we find

$$f(x_k) - f(x^*) \le \frac{f(x_0) - f(x^*)}{(1 + \delta k/(ap))^p} + w_a(\delta)^{p-1} c_p \frac{R^p}{a^p} \frac{\delta k}{(1 + \delta k/(ap))^p}.$$

Since $a > 0$ was arbitrary, we may choose $a = b w_b(\delta)^{(p-1)/p}$ for $b = R \frac{(c_p \delta k)^{1/p}}{(f(x_0) - f(x^*))^{1/p}}$. Since $w_b(\delta) \ge 1$, we have $b \le a$ and hence $w_a(\delta) \le w_b(\delta)$. Therefore,

$$f(x_k) - f(x^*) \le \frac{2(f(x_0) - f(x^*))}{\left(1 + \frac{(f(x_0) - f(x^*))^{1/p}}{R c_p^{1/p} p w_b(\delta)^{(p-1)/p}} (\delta k)^{\frac{p-1}{p}}\right)^p} = O(1/(1 + \frac{1}{Rp}(\delta k)^{\frac{p-1}{p}})^p)$$

as desired.

### B.1.3 Proof of Theorem 3

Take the energy function $E_k = f(x_k) - f(x^*)$. Observe that if (2a) holds, then we have:

$$\frac{E_{k+1} - E_k}{\delta} = \frac{f(x_{k+1}) - f(x_k)}{\delta} \overset{(2a)}{\le} -\|\nabla f(x_k)\|_*^{\frac{p}{p-1}} \overset{(3)}{\le} -\frac{p}{p-1}\mu^{\frac{1}{p-1}} E_k,$$

or rewritten, $E_{k+1} \le \left(1 - \frac{p}{p-1}\mu^{\frac{1}{p-1}}\delta\right) E_k$. Summing gives the bound

$$E_{k+1} \le \left(1 - \frac{p}{p-1}\mu^{\frac{1}{p-1}}\delta\right)^k E_0 \le e^{-\frac{p}{p-1}\mu^{\frac{1}{p-1}}\delta k} E_0,$$

using $1 + x \le e^x \ \forall x \in \mathbb{R}$. On the other hand, if (2b) holds, then a similar argument follows:

$$\frac{E_{k+1} - E_k}{\delta} = \frac{f(x_{k+1}) - f(x_k)}{\delta} \overset{(2b)}{\le} -\|\nabla f(x_{k+1})\|_*^{\frac{p}{p-1}} \overset{(3)}{\le} -\frac{p}{p-1}\mu^{\frac{1}{p-1}} E_{k+1},$$

or rewritten, $E_{k+1} \le \left(1 + \frac{p}{p-1}\mu^{\frac{1}{p-1}}\delta\right)^{-1} E_k$. Summing gives the bound

$$E_{k+1} \le \left(1 + \frac{p}{p-1}\mu^{\frac{1}{p-1}}\delta\right)^{-k} E_0 \le e^{-\frac{p}{p-1}\mu^{\frac{1}{p-1}}\delta k} E_0.$$

## B.2 Examples of descent methods

We now provide detailed demonstration that the examples provided are descent algorithms.

### B.2.1 Higher-order gradient descent

Let $\tilde{p} = p - 1 + \nu$. The optimality condition for the HGD algorithm (7) is

$$\sum_{i=1}^{p-1} \frac{1}{(i-1)!} \nabla^i f(x_k)(x_{k+1} - x_k)^{i-1} + \frac{1}{\eta}\|x_{k+1} - x_k\|^{\tilde{p}-2} B(x_{k+1} - x_k) = 0. \qquad (34)$$

Since $\nabla^{p-1} f$ is $L$-Lipschitz, we have the following error bound on the $(p-2)$-nd order Taylor expansion of $\nabla f$:

$$\left\|\nabla f(x_{k+1}) - \sum_{i=1}^{p-1} \frac{1}{(i-1)!} \nabla^i f(x_k)(x_{k+1} - x_k)^{i-1}\right\|_* \le \frac{L}{(p-2)!}\|x_{k+1} - x_k\|^{p-2+\nu}. \qquad (35)$$

Substituting (34) to (35) and writing $r_k = \|x_{k+1} - x_k\|$, we obtain

$$\left\|\nabla f(x_{k+1}) + \frac{r_k^{\tilde{p}-2}}{\eta} B(x_{k+1} - x_k)\right\|_* \le \frac{L}{(p-2)!} r_k^{\tilde{p}-1}. \qquad (36)$$

Squaring both sides, expanding, and rearranging the terms, we get the inequality

$$\langle \nabla f(x_{k+1}), x_k - x_{k+1} \rangle \ge \frac{\eta}{2 r_k^{\tilde{p}-2}} \|\nabla f(x_{k+1})\|_*^2 + \frac{\eta r_k^{\tilde{p}}}{2}\left(\frac{1}{\eta^2} - \frac{L^2}{(p-2)!^2}\right). \qquad (37)$$

If $p = 2$, then the first term in (37) already implies the desired bound below. Now assume $p \ge 3$. The right-hand side of (37) is of the form $A/r^{\tilde{p}-2} + B r^{\tilde{p}}$, which is a convex function of $r > 0$ and minimized by $r^* = \left\{\frac{(\tilde{p}-2)}{\tilde{p}}\frac{A}{B}\right\}^{\frac{1}{2\tilde{p}-2}}$, yielding a minimum value of

$$\frac{A}{(r^*)^{\tilde{p}-2}} + B(r^*)^{\tilde{p}} = A^{\frac{\tilde{p}}{2\tilde{p}-2}} B^{\frac{\tilde{p}-2}{2\tilde{p}-2}}\left[\left(\frac{\tilde{p}}{\tilde{p}-2}\right)^{\frac{\tilde{p}-2}{2\tilde{p}-2}} + \left(\frac{\tilde{p}-2}{\tilde{p}}\right)^{\frac{\tilde{p}}{2\tilde{p}-2}}\right] \ge A^{\frac{\tilde{p}}{2\tilde{p}-2}} B^{\frac{\tilde{p}-2}{2\tilde{p}-2}}.$$

Substituting the values $A = \frac{\eta}{2}\|\nabla f(x_{k+1})\|_*^2$ and $B = \frac{\eta}{2}(\frac{1}{\eta^2} - \frac{L^2}{(p-2)!^2})$ from (37), we obtain

$$\langle \nabla f(x_{k+1}), x_k - x_{k+1} \rangle \geq \frac{\eta}{2}\left(\frac{1}{\eta^2} - \frac{L^2}{(p-2)!^2}\right)^{\frac{\tilde{p}-2}{2\tilde{p}-2}} \|\nabla f(x_{k+1})\|_*^{\frac{\tilde{p}}{\tilde{p}-1}}.$$

Finally, using the inequality $f(x_k) - f(x_{k+1}) \geq \langle \nabla f(x_{k+1}), x_k - x_{k+1} \rangle$ by the convexity of $f$ yields the progress bound

$$f(x_{k+1}) - f(x_k) \leq -\frac{\eta^{\frac{1}{\tilde{p}-1}}}{2}\left(1 - \frac{(L\eta)^2}{(p-2)!^2}\right)^{\frac{\tilde{p}-2}{2\tilde{p}-2}} \|\nabla f(x_{k+1})\|_*^{\frac{\tilde{p}}{\tilde{p}-1}}$$

$$\leq -\frac{\eta^{\frac{1}{\tilde{p}-1}}}{2^{\frac{2\tilde{p}-3}{\tilde{p}-1}}} \|\nabla f(x_{k+1})\|_*^{\frac{\tilde{p}}{\tilde{p}-1}}$$

where the least inequality uses the fact that $\eta \leq \frac{\sqrt{3}(p-2)!}{2L}$.

### B.2.2 Proximal method

The optimality condition for the proximal method is

$$\nabla^2 h(x_k)^{-1}\nabla f(x_{k+1}) + \frac{\|x_{k+1} - x_k\|_{x_k}^{p-2}}{\eta}(x_{k+1} - x_k) = 0,$$

which implies $\|x_{k+1} - x_k\|_{x_k} = \eta^{\frac{1}{p-1}}\|\nabla f(x_{k+1})\|_{x_k,*}^{\frac{1}{p-1}}$, using the shorthand $\|v\|_{x_k,*} = \sqrt{\langle v, \nabla h(x_k)^{-1} v \rangle}$. From the definition of $x_{k+1}$, we have $f(x_{k+1}) + \frac{1}{p\eta}\|x_{k+1} - x_k\|_{x_k}^p \leq f(x_k)$. Rearranging gives

$$f(x_k) - f(x_{k+1}) \geq \frac{1}{p\eta}\|x_{k+1} - x_k\|_{x_k}^p = \frac{\eta^{\frac{1}{p-1}}}{p}\|\nabla f(x_{k+1})\|_{*,x_k}^{\frac{p}{p-1}} \geq \frac{m^{\frac{p}{p-1}}\eta^{\frac{1}{p-1}}}{p}\|\nabla f(x_{k+1})\|_*^{\frac{p}{p-1}}$$

as desired.

### B.2.3 Natural gradient descent

Since $\nabla^2 f \preceq LB$, we have the bound

$$f(x_{k+1}) \leq f(x_k) + \langle \nabla f(x_k), x_{k+1} - x_k \rangle + \frac{L}{2}\|x_{k+1} - x_k\|^2.$$

Plugging in the NGD update (9) gives

$$f(x_{k+1}) \leq f(x_k) - \eta\langle \nabla f(x_k), (\nabla^2 h(x_k))^{-1}\nabla f(x_k)\rangle + \frac{L\eta^2}{2}\langle \nabla f(x_k), B(\nabla^2 h(x_k))^{-2}\nabla f(x_k)\rangle.$$

Since $mB \preceq \nabla^2 h \preceq MB$, we have $\frac{1}{M}B^{-1} \preceq (\nabla^2 h)^{-1} \preceq \frac{1}{m}B^{-1}$, so

$$f(x_{k+1}) \leq f(x_k) - \frac{\eta}{M}\|\nabla f(x_k)\|_*^2 + \frac{L\eta^2}{2m^2}\|\nabla f(x_k)\|_*^2$$

$$= f(x_k) - \eta\left(\frac{1}{M} - \frac{L\eta}{2m^2}\right)\|\nabla f(x_k)\|_*^2$$

$$\leq f(x_k) - \frac{\eta}{2M}\|\nabla f(x_k)\|_*^2$$

where in the last step we have used the inequality $\eta \leq \frac{m^2}{ML}$.

### B.2.4 Mirror descent

Plugging the variational condition $\nabla h(x_{k+1}) - \nabla h(x_k) = -\eta\nabla f(x_k)$ into the smoothness bound on $f$, as well as using the property $mB \preceq \nabla^2 h$ we have

$$f(x_{k+1}) - f(x_k) \leq \langle \nabla f(x_k), x_{k+1} - x_k \rangle + \frac{L}{2}\|x_{k+1} - x_k\|^2$$

$$\leq -\frac{1}{\eta}\langle \nabla h(x_{k+1}) - \nabla h(x_k), x_{k+1} - x_k \rangle + \frac{L}{2m^2}\|\nabla h(x_{k+1}) - \nabla h(x_k)\|_*^2$$

Given $h$ is $M$-smooth, $-\frac{1}{\eta}\langle \nabla h(x_{k+1}) - \nabla h(x_k), x_{k+1} - x_k \rangle \leq -\frac{1}{\eta M}\|\nabla h(x_{k+1}) - \nabla h(x_k)\|_*^2$ ((Nesterov, 2004, (2.1.8))) and therefore,

$$f(x_{k+1}) - f(x_k) \leq -\left(\frac{1}{\eta M} - \frac{L}{2m^2}\right)\|\nabla h(x_{k+1}) - \nabla h(x_k)\|_*^2 \leq -\eta\left(\frac{1}{M} - \frac{L\eta}{2m^2}\right)\|\nabla f(x_k)\|_*^2$$

$$\leq -\frac{\eta}{2M}\|\nabla f(x_k)\|_*^2$$

where in the last step we have used the inequality $\eta \leq \frac{m^2}{ML}$.

### B.2.5 Proximal Bregman Method

The optimality condition for the proximal method is $\eta \nabla f(x_{k+1}) = \nabla h(x_{k+1}) - \nabla h(x_k)$, which implies $\eta^2 \|\nabla f(x_{k+1})\|_*^2 = \|\nabla h(x_{k+1}) - \nabla h(x_k)\|_*^2 \le M^2 \|x_{k+1} - x_k\|^2$. From the definition of $x_{k+1}$, we have $f(x_{k+1}) + \frac{1}{\eta} D_h(x_{k+1}, x_k) \le f(x_k)$. Rearranging gives

$$f(x_{k+1}) - f(x_k) \le -\frac{1}{\eta} D_h(x_{k+1}, x_k) \le -\frac{m}{2\eta} \|x_{k+1} - x_k\|^2 \le -\frac{m\eta}{2M^2} \|\nabla f(x_{k+1})\|_*^2$$

as desired.

### B.3 Rescaled Gradient Descent

**Proof of Lemma 4** We show rescaled gradient descent satisfies progress bound (2) with $\delta = \eta^{\frac{1}{p-1}}/2$ when $f$ is strongly smooth. Since $\|\nabla^p f(x)\| \le L_p$, we have the Taylor expansion bound,

$$
\begin{aligned}
f(x_{k+1}) - f(x_k) &\le \langle \nabla f(x_k), x_{k+1} - x_k \rangle + \sum_{m=2}^{p-1} \frac{1}{m!} \nabla^m f(x_k)(x_{k+1} - x_k)^m + \frac{L_p}{p!} \|x_{k+1} - x_k\|^p \\
&\overset{(12)}{=} -\eta^{\frac{1}{p-1}} \left(1 - \frac{\eta L_p}{p!}\right) \|\nabla f(x_k)\|_*^{\frac{p}{p-1}} + \sum_{m=2}^{p-1} \frac{\eta^{\frac{m}{p-1}}}{m!} \frac{\nabla^m f(x_k)(\nabla f(x_k))^m}{\|\nabla f(x_k)\|_*^{\frac{m(p-2)}{p-1}}} \\
&\overset{(13)}{\le} -\eta^{\frac{1}{p-1}} \left(1 - \frac{\eta L_p}{p!}\right) \|\nabla f(x_k)\|_*^{\frac{p}{p-1}} + \sum_{m=2}^{p-1} \frac{\eta^{\frac{m}{p-1}}}{m!} L_m \|\nabla f(x_k)\|_*^{m + \frac{p-m}{p-1} - \frac{m(p-2)}{p-1}} \\
&= -\eta^{\frac{1}{p-1}} \left(1 - \frac{\eta L_p}{p!}\right) \|\nabla f(x_k)\|_*^{\frac{p}{p-1}} + \sum_{m=2}^{p-1} \frac{\eta^{\frac{m}{p-1}}}{m!} L_m \|\nabla f(x_k)\|_*^{\frac{p}{p-1}} \\
&= -\eta^{\frac{1}{p-1}} \left(1 - \sum_{m=2}^{p} \frac{\eta^{\frac{m-1}{p-1}} L_m}{m!}\right) \|\nabla f(x_k)\|_*^{\frac{p}{p-1}}.
\end{aligned}
$$

The second line follows from the rescaled gradient update (12) and the third follows from our strongly smoothness Assumption (def 2). Since $\eta < 1$ we can further bound

$$f(x_{k+1}) - f(x_k) \le -\eta^{\frac{1}{p-1}} \left(1 - \eta^{\frac{1}{p-1}} \sum_{m=2}^{p} \frac{L_m}{m!}\right) \|\nabla f(x_k)\|_*^{\frac{p}{p-1}}.$$

Our step-size condition (14) implies $1 - \eta^{\frac{1}{p-1}} \sum_{m=2}^{p} \frac{L_m}{m!} \ge \frac{1}{2}$, which yields the desired bound (2) with $\delta = \eta^{\frac{1}{p-1}}/2$.

### B.4 Gradient Descent vs. Rescaled Gradient Descent

**Proof of Lemma 4** We have $f'(x) = \text{sign}(x)|x|^{p-1}$, so $|f'(x)|^{\frac{p-2}{p-1}} = |x|^{p-2}$.

The rescaled gradient descent of order $p$ with step size $\epsilon = \eta^{\frac{1}{p-1}}$ is

$$x_{k+1} = x_k - \epsilon \frac{f'(x_k)}{|f'(x_k)|^{\frac{p-2}{p-1}}} = x_k - \epsilon \frac{\text{sign}(x_k)|x_k|^{p-1}}{|x_k|^{p-2}} = (1 - \epsilon)x_k.$$

Therefore, if $0 < \epsilon < 1$, then $x_k = (1 - \epsilon)^k x_0$, and thus $f(x_k) = (1 - \epsilon)^{pk} f(x_0)$ converges to 0 at an exponential rate $\Theta((1 - \epsilon)^{pk})$.

The gradient descent with step size $\epsilon = \eta^{\frac{1}{p-1}}$ for $f$ is

$$x_{k+1} = x_k - \epsilon f'(x_k) = x_k - \epsilon \, \text{sign}(x_k)|x_k|^{p-1} = (1 - \epsilon |x_k|^{p-2})x_k.$$

Note that if $0 < \epsilon < |x_k|^{-(p-2)}$, then $x_{k+1}$ has the same sign as $x_k$ with smaller magnitude. In particular, if $0 < x_0 < \epsilon^{-\frac{1}{p-2}}$, then $x_k > x_{k+1} > 0$ for all $k > 0$, and gradient descent simplifies to $x_{k+1} = (1 - \epsilon x_k^{p-2})x_k$. Assume we start with $0 < x_0 \le (2\epsilon)^{-\frac{1}{p-2}}$, so $\frac{x_k}{x_{k+1}} = (1 - \epsilon x_k^{p-2})^{-1} \le (1 - \epsilon x_0^{p-2})^{-1} \le 2$. Then by Jensen's inequality applied to the convex function $x \mapsto x^{-(p-2)}$, we have $x_{k+1}^{-(p-2)} - x_k^{-(p-2)} \le \frac{-(p-2)}{x_{k+1}^{p-1}}(x_k - x_{k+1}) = (p-2)\epsilon \frac{x_k^{p-1}}{x_{k+1}^{p-1}} \le (p-2)2^{p-1}\epsilon$. This implies $x_k \ge (x_0^{-(p-2)} + (p-2)2^{p-1}\epsilon k)^{-\frac{1}{p-2}} = \Omega((\epsilon k)^{-\frac{1}{p-2}})$, and thus $f(x_k) \ge \Omega((\epsilon k)^{-\frac{p}{p-2}})$ converges to 0 at a polynomial rate.

### B.4.1 Gradient Flow vs. Rescaled Gradient Flow

We also discuss how the behavior in discrete time above matches the behavior in continuous time. The rescaled gradient flow of order $p$ for $f$ is

$$\dot{X}_t = -\frac{f'(X_t)}{|f'(X_t)|^{\frac{p-2}{p-1}}} = -\frac{\text{sign}(X_t)|X_t|^{p-1}}{|X_t|^{p-2}} = -X_t$$

so $X_t = e^{-t}X_0$, and thus $f(X_t) = e^{-pt}f(X_0)$ converges to 0 at an exponential rate $\Theta(e^{-pt})$.

The gradient flow (which is rescaled gradient flow of order 2) for $f$ is

$$\dot{X}_t = -f'(X_t) = -\text{sign}(X_t)|X_t|^{p-1}$$

Without loss of generality assume $X_0 > 0$, so $X_t > 0$ for all $t > 0$. Then gradient flow simplifies to $\dot{X}_t = -X_t^{p-1}$, or $\frac{d}{dt}X_t^{-(p-2)} = -(p-2)\dot{X}_t X_t^{-(p-1)} = p-2$, so $X_t = (X_0^{-(p-2)} + (p-2)t)^{-\frac{1}{p-2}}$, and thus $f(X_t) = \Theta(t^{-\frac{p}{p-2}})$ converges to 0 at a polynomial rate.

More generally, the rescaled gradient flow of order $q$ ($q > 1, q \neq p$) for $f$ is

$$\dot{X}_t = -\frac{f'(X_t)}{|f'(X_t)|^{\frac{q-2}{q-1}}} = -\frac{\text{sign}(X_t)|X_t|^{p-1}}{|X_t|^{\frac{(q-2)(p-1)}{q-1}}} = -\text{sign}(X_t)|X_t|^{\frac{p-1}{q-1}}$$

Assume $X_0 > 0$, so $X_t > 0$ for all $t > 0$. Rescaled gradient flow simplifies to $\dot{X}_t = -X_t^{\frac{p-1}{q-1}}$, or $\frac{d}{dt}X_t^{-\frac{p-q}{q-1}} = \frac{p-q}{q-1}$, so $X_t = (X_0^{-\frac{p-q}{q-1}} + (\frac{p-q}{q-1})t)^{-\frac{q-1}{p-q}}$, and $f(X_t) = \Theta(t^{-\frac{p(q-1)}{p-q}})$. Note that if $1 < q < p$, then $f(X_t)$ converges to 0 at a polynomial rate, which becomes faster as $q \to p$. At $q = p$, the convergence rate becomes exponential, as we see for rescaled gradient flow above. However, for $q > p$, $f(X_t)$ diverges to $\infty$. Thus, the best order to use is $q = p$, but it is better to underestimate $p$.

## C  Accelerating Descent Algorithms

The energy function

$$E_k = D_h(x^*, z_k) + A_k(f(y_k) - f(x^*)), \tag{38}$$

will be used to analyze all the accelerated methods introduced in this paper.

### C.1  Proof of Proposition 7

Take energy (Lyapunov) function (38) Set $A_k = C\delta^p k^{(p)}$ where $k^{(p)} = k(k+1)\cdots(k+p-1)$ is the rising factorial. Denote $\alpha_k := \frac{A_{k+1}-A_k}{\delta} = Cp\delta^{p-1}(k+1)^{(p-1)}$ and $\tau_k := \frac{\alpha_k}{A_{k+1}} = \frac{k}{\delta(k+p)}$.

**Algorithm** (15):  Using (38) we compute

$$\frac{E_{k+1}-E_k}{\delta} = \frac{D_h(x^*,z_{k+1})-D_h(x^*,z_k)}{\delta} + \frac{A_{k+1}}{\delta}(f(y_{k+1}) - f(x^*)) - \frac{A_k}{\delta}(f(y_k) - f(x^*)). \tag{39}$$

We bound the first part,

$$\frac{D_h(x^*,z_{k+1})-D_h(x^*,z_k)}{\delta} = -\left\langle \frac{\nabla h(z_{k+1})-\nabla h(z_k)}{\delta}, x^* - z_{k+1} \right\rangle - \frac{1}{\delta}D_h(z_{k+1},z_k)$$

$$\stackrel{(15b)}{=} \alpha_k\langle \nabla f(x_k), x^* - z_k\rangle + \alpha_k\langle \nabla f(x_k), z_k - z_{k+1}\rangle - \frac{1}{\delta}D_h(z_{k+1},z_k)$$

$$\leq \alpha_k\langle \nabla f(x_k), x^* - z_k\rangle + (\delta/m)^{\frac{1}{p-1}}\alpha_k^{\frac{p}{p-1}}\|\nabla f(x_k)\|_*^{\frac{p}{p-1}}, \tag{40}$$

where the inequality follows from the $m$-uniform convexity of $h$ of order $p$ and the Fenchel-Young inequality $\langle s, h\rangle + \frac{1}{p}\|h\|^p \geq -\frac{p}{p-1}\|s\|_*^{\frac{p}{p-1}} \leq -\|s\|_*^{\frac{p}{p-1}}$, with $h = (m/\delta)^{\frac{1}{p}}(z_{k+1} - z_k)$ and $s = (\delta/m)^{\frac{1}{p}}\alpha_k^{\frac{p}{p-1}}\nabla f(x_k)$. Plugging in update (15a),

$$\alpha_k\langle \nabla f(x_k), x^* - z_k\rangle = \alpha_k\langle \nabla f(x_k), x^* - y_k\rangle + \frac{A_{k+1}}{\delta}\langle \nabla f(x_k), y_k - x_k\rangle$$

$$= \alpha_k \langle \nabla f(x_k), x^* - x_k \rangle + \tfrac{A_k}{\delta} \langle \nabla f(x_k), y_k - x_k \rangle$$
$$\leq - \left( \tfrac{A_{k+1}}{\delta} (f(y_{k+1}) - f(x^*)) - \tfrac{A_k}{\delta} (f(y_k) - f(x^*)) \right)$$
$$+ A_{k+1} \tfrac{f(y_{k+1}) - f(x_k)}{\delta}$$
$$\leq - \left( \tfrac{A_{k+1}}{\delta} (f(y_{k+1}) - f(x^*)) - \tfrac{A_k}{\delta} (f(y_k) - f(x^*)) \right)$$
$$- A_{k+1} \delta^{\frac{1}{p-1}} \| \nabla f(x_k) \|_*^{\frac{p}{p-1}}. \tag{41}$$

The first inequality follows from the convexity of $f$ and rearranging terms. The second inequality uses the progress condition assumed for the sequence $y_{k+1}$. Combining (39) with (40) and (41) we have,

$$\tfrac{E_{k+1} - E_k}{\delta} \leq \left( (\delta/m)^{\frac{1}{p-1}} (Cp\delta^{p-1}(k+1)^{(p-1)})^{\frac{p}{p-1}} - C\delta^{\frac{1}{p-1}} \delta^p (k+1)^{(p)} \right) \| \nabla f(x_k) \|_*^{\frac{p}{p-1}}.$$

Given $((k+1)^{(p-1)})^{\frac{p}{p-1}} / (k+1)^{(p)} \leq 1$, it suffices that $C \leq 1/mp^p$ to ensure $\tfrac{E_{k+1} - E_k}{\delta} \leq 0$. Summing the Lyapunov function gives the convergence rate $f(y_k) - f(x^*) = O(1/A_k) = O(1/(\delta k)^p)$.

**Algorithm (16):** Using (38) with the same parameter choices as algorithm (15), we have

$$\tfrac{D_h(x^*, z_{k+1}) - D_h(x^*, z_k)}{\delta} \leq \alpha_k \langle \nabla f(y_{k+1}), x^* - z_k \rangle + (\delta/m)^{\frac{1}{p-1}} \alpha_k^{\frac{p}{p-1}} \| \nabla f(y_{k+1}) \|_*^{\frac{p}{p-1}}, \tag{42}$$

where the first part uses the same steps as (40) except update (16b) is used instead of (15b). Plugging in update (16a) yields the following,

$$\alpha_k \langle \nabla f(y_{k+1}), x^* - z_k \rangle = \alpha_k \langle \nabla f(y_{k+1}), x^* - y_{k+1} \rangle + \tfrac{A_{k+1}}{\delta} \langle \nabla f(y_{k+1}), y_{k+1} - z_k \rangle$$
$$\stackrel{(16a)}{=} \alpha_k \langle \nabla f(y_{k+1}), x^* - y_{k+1} \rangle + \tfrac{A_k}{\delta} \langle \nabla f(y_{k+1}), y_k - y_{k+1} \rangle$$
$$+ \tfrac{A_{k+1}}{\delta} \langle \nabla f(y_{k+1}), y_{k+1} - x_k \rangle$$
$$\leq - \left( \tfrac{A_{k+1}}{\delta} (f(y_{k+1}) - f(x^*)) - \tfrac{A_k}{\delta} (f(y_k) - f(x^*)) \right)$$
$$+ \tfrac{A_{k+1}}{\delta} \langle \nabla f(y_{k+1}), y_{k+1} - x_k \rangle$$
$$\leq - \left( \tfrac{A_{k+1}}{\delta} (f(y_{k+1}) - f(x^*)) - \tfrac{A_k}{\delta} (f(y_k) - f(x^*)) \right)$$
$$- A_{k+1} \delta^{\frac{1}{p-1}} \| \nabla f(y_{k+1}) \|_*^{\frac{p}{p-1}}. \tag{43}$$

The first inequality follows from the convexity of $f$ and rearranging terms. The second inequality uses the progress condition assumed for the sequence $y_{k+1}$. Combining (39) with (42) (43), we have

$$\tfrac{E_{k+1} - E_k}{\delta} \leq -\delta^{\frac{1}{p-1}} C(k+1)^{(p)} \| \nabla f(y_{k+1}) \|_*^{\frac{p}{p-1}} + (\delta/m)^{\frac{1}{p-1}} (Cp(k+1)^{(p-1)})^{\frac{p}{p-1}} \| \nabla f(y_{k+1}) \|_*^{\frac{p}{p-1}}.$$

For $\tfrac{E_{k+1} - E_k}{\delta} \leq 0$ it suffices that $C \leq 1/mp^p$. Summing the Lyapunov function gives the convergence rate $f(y_k) - f(x^*) = O(1/A_k) = O(1/(\delta k)^p)$.

## C.2 Restarting Scheme

When $f$ is *strongly smooth* and $\mu$-gradient dominated, we define the restarting scheme (similar to (Wibisono et al., 2016, (B.1.2))), which proceeds by running 1 for some number of iterations at each step,

$$\hat{x}_k = \text{(the output } y_c \text{ of running Algorithm 1 for } c \text{ iterations with input } x_0 = \hat{x}_{k-c}). \tag{44}$$

**Theorem 14** *Assume $f$ is convex and strongly smooth of order $1 < p < \infty$ with constants $0 < L_1, \ldots, L_p < \infty$ and $f$ is $\mu$-gradient dominated of order $p$. Suppose $\eta$ satisfies (14). Let $\hat{x}_k$ be the output of running the restarting scheme (44) for $k/c$ times with $c = 2p/\kappa^{\frac{1}{p}}$ where $\kappa = \mu\delta^p = \mu\eta$. Finally, let $y_k$ be the output of running the rescaled gradient descent update one step from $\hat{x}_k$. The composite scheme satisfies the convergence rate upper bound: $f(y_k) - f(x^*) = O(\exp(-\frac{1}{2p} \mu^{\frac{1}{p}} \delta k)$*

Take $h(x) = \frac{2^{p-2}}{p}\|x - x_0\|^p$ which is 1-uniformly convex of order $p$. Running $k$ iterations of either algorithm (15) or (16) results in the convergence bound,

$$\frac{\mu}{p}\|\hat{x}_k - x^*\|^p \leq f(\hat{x}_k) - f(x^*) \leq \frac{2^{p-2}p^{p-1}\|\hat{x}_{k-c} - x^*\|^p}{\delta^p k^p} \leq \frac{2^{p-2}p^{p-1}\|\hat{x}_{k-c} - x^*\|^p}{(\delta c)^p}$$
$$\leq \frac{\mu}{pe}\|\hat{x}_{k-c} - x^*\|^p. \qquad (45)$$

where the last inequality follows from the choice $c = 2p/\kappa^{\frac{1}{p}}$. Thus an execution of (44) for $c$ iterations of the accelerated method reduces the distance to optimum by a factor of at least $1/e$. Iterating (45), we obtain $\frac{1}{p}\|\hat{x}_k - x^*\|^p \leq e^{-k/c}\frac{1}{p}\|\hat{x}_0 - x^*\|^p$. Using the descent property for both methods, $E_{k+1} \leq \delta 2p^{p-1}\|x_k - x^*\|^p$ (2a) and $E_{k+1} \leq \delta 2p^{p-1}\|x_{k+1} - x^*\|^p$ (2b), implies that

$$f(\hat{y}_k) - f(x^*) \leq \delta 2p^{p-1}e^{\frac{-\kappa^{\frac{1}{p}}k}{2p}}\|x_0 - x^*\|^p = O\left(e^{\frac{-\kappa^{\frac{1}{p}}k}{2p}}\right).$$

## C.3  Proof of Proposition 9

We analyze the following sequence of iterates

$$x_k = \delta\tau_k z_k + (1 - \delta\tau_k)y_k \qquad (46a)$$
$$z_{k+1} = \arg\min_z \left\{\alpha_k\langle\nabla f(y_{k+1}), z\rangle + \frac{1}{\delta}D_h(z, z_k)\right\}, \qquad (46b)$$

where the update for $(\lambda_{k+1}, y_{k+1})$ satisfies the descent conditions

$$a \leq \frac{\lambda_{k+1}}{\delta^{\frac{3p-2}{2}}}\|y_{k+1} - x_k\|^{p-2} \leq b, \qquad (46c)$$

$$\|y_{k+1} - x_k + \frac{\lambda_{k+1}}{m}\nabla f(y_{k+1})\| \leq \frac{1}{2}\|y_{k+1} - x_k\|, \qquad (46d)$$

and the following identifications $\alpha_k = \frac{A_{k+1} - A_k}{\delta}$, $\tau_k = \frac{\alpha_k}{A_{k+1}}$, and $\lambda_{k+1} = \frac{\alpha_k^2}{\delta^2 A_{k+1}}$ hold. Assume $h$ is $m$-strongly convex.

Taking energy function (38), we compute

$$\frac{E_{k+1} - E_k}{\delta} = \frac{A_{k+1}}{\delta}(f(y_{k+1}) - f(x^*)) - \frac{A_k}{\delta}(f(y_k) - f(x^*))$$
$$- \left\langle\frac{\nabla h(z_{k+1}) - \nabla h(z_k)}{\delta}, x^* - z_{k+1}\right\rangle - D_h(z_{k+1}, z_k)$$
$$\overset{(46b)}{\leq} \alpha_k(f(y_{k+1}) - f(x^*)) + \frac{A_k}{\delta}(f(y_{k+1}) - f(y_k)) + \alpha_k\langle\nabla f(y_{k+1}), x^* - z_{k+1}\rangle$$
$$- \frac{m}{2\delta}\|z_k - z_{k+1}\|^2$$
$$\leq \alpha_k\langle\nabla f(y_{k+1}), y_{k+1} - z_{k+1}\rangle + \frac{A_k}{\delta}\langle\nabla f(y_{k+1}), y_k - y_{k+1}\rangle - \frac{m}{2\delta}\|z_k - z_{k+1}\|^2.$$

where the first inequality follows from the strong convexity of $h$ and the last inequality follows from the convexity of $f$. Denote $x = \delta\tau_k z_{k+1} + (1 - \delta\tau_k)y_k$. Starting from the preceding line, we have,

$$\frac{E_{k+1} - E_k}{\delta} \leq \frac{A_{k+1}}{\delta}\langle\nabla f(y_{k+1}), y_{k+1} - x\rangle - \frac{m}{2\delta}\|z_k - \frac{1}{\delta\tau_k}x + \frac{1-\delta\tau_k}{\delta\tau_k}y_k\|^2$$
$$= \frac{A_{k+1}}{\delta}\langle\nabla f(y_{k+1}), y_{k+1} - x\rangle - \frac{1}{2(\delta\tau_k)^2}\frac{m}{\delta}\|\delta\tau_k z_k + (1-\delta\tau_k)y_k - x\|^2$$
$$\overset{(46a)}{=} \frac{A_{k+1}}{\delta}\langle\nabla f(y_{k+1}), y_{k+1} - x\rangle - \frac{1}{2(\delta\tau_k)^2}\frac{m}{\delta}\|x_k - x\|^2$$
$$\leq \max_{x\in\mathcal{X}}\left\{\frac{A_{k+1}}{\delta}\langle\nabla f(y_{k+1}), y_{k+1} - x\rangle - \frac{1}{2(\delta\tau_k)^2}\frac{m}{\delta}\|x_k - x\|^2\right\}.$$

Plugging in the solution, which satisfies $x = x_k - \frac{\delta^2}{m}\frac{\alpha_k^2}{A_{k+1}}\nabla f(y_{k+1})$, and noting $\lambda_{k+1} = \frac{\delta^2\alpha_k^2}{A_{k+1}}$ we obtain

$$\frac{E_{k+1} - E_k}{\delta} \leq \frac{A_{k+1}}{\lambda_{k+1}}\frac{m}{2\delta}\left(\|y_{k+1} - x_k + \frac{\lambda_{k+1}}{m}\nabla f(y_{k+1})\|^2 - \|y_{k+1} - x_k\|^2\right)$$
$$\overset{(46d)}{\leq} -\frac{A_{k+1}}{\lambda_{k+1}\delta}\frac{m}{4}\|y_{k+1} - x_k\|^2. \qquad (47)$$

This is the same bound as (Monteiro and Svaiter, 2013, (3.12)) with $\sigma = 0$.

Rearranging the last inequality and summing over $k$, we have

$$\sum_{i=0}^{k} \frac{A_i}{\lambda_i} \frac{m}{4} \|y_{i+1} - x_i\|^2 \leq E_{k+1} + \sum_{i=0}^{k} \frac{A_i}{\lambda_i} \frac{m}{4} \|y_{i+1} - x_i\|^2 \leq E_0 = D_h(x^*, x_0), \qquad (48)$$

where the last equality comes from taking $A_0 = 0$.

Notice that summing over our bound (47) gives us the rate

$$f(y_k) - f(x^*) \leq \frac{E_0}{A_k}.$$

Now we use the second bound (46c) to establish $A_k = O(k^{\frac{3p-2}{2}})$. This follows from arguments identical to the those given by (Gasnikov et al., 2019, p.6-7) and (Bubeck et al., p.6-8). Denote $a_1 = a\delta^{\frac{3p-2}{2}}$. Observe that

$$\sum_{i=0}^{k} \frac{A_i}{\lambda_i^{\frac{p}{p-2}}} a_1^{\frac{2}{p-2}} \overset{(46c)}{\leq} \sum_{i=0}^{k} \frac{A_i}{\lambda_i^{1+\frac{2}{p-2}}} \left(\lambda_i \|y_{i+1} - x_i\|^{p-2}\right)^{\frac{2}{p-2}} \leq \sum_{i=0}^{k} \frac{A_i}{\lambda_i} \|y_{i+1} - x_i\|^2 \overset{(48)}{\leq} 4E_0/m. \tag{49}$$

Denote $c_1 = a_1^{-\frac{2}{p-2}} 4E_0/m = (a\delta^{\frac{3p-2}{2}})^{-\frac{2}{p-2}} E_0 4/m$. Using the previous line, we have

$$A_k \geq \frac{1}{4} \left(\sum_{i=1}^{k} \sqrt{\lambda_i}\right)^2 \geq \frac{1}{4} c_1^{-\frac{p-2}{p}} \left(\sum_{i=1}^{k} A_i^{\frac{p-2}{3p-2}}\right)^{\frac{3p-2}{p}}, \tag{50}$$

where the first inequality follows from definition of $\alpha_k$ (see (Bubeck et al., Lem 2.6)) and the second inequality uses reverse Holders (see (Bubeck et al., p.7-8)). Specifically, we have

$$\alpha_k = \frac{\lambda_k + \sqrt{\lambda_k^2 + 4\lambda_k A_{k-1}}}{2} \geq \frac{\lambda_k}{2} + \sqrt{\lambda_k A_{k-1}} \geq \left(\frac{\lambda_k}{2} + \sqrt{A_{k-1}}\right)^2 - A_{k-1},$$

and $\alpha_k^2 = \lambda_k A_k$ which allows us to conclude the first inequality. For the second inequality, we use reverse Holder (i.e. $\|fg\|_1 \geq \|f\|_{\frac{1}{q}} \|g\|_{-\frac{1}{q-1}}$ for $q \geq 1$) with $q = 1 + \frac{p-2}{2p} = \frac{3p-2}{2p}$ so that $-\frac{1}{q-1} = \frac{2p}{p-2}$, we have

$$\sum_{i=0}^{k} \sqrt{\lambda_i} = \sum_{i=0}^{k} A_i^{\frac{p-2}{2p}} \left(\frac{A_i}{\lambda_i^{\frac{p}{p-2}}}\right)^{-\frac{p-2}{2p}} \geq \left(\sum_{i=0}^{k} A_i^{\frac{p-2}{3p-2}}\right)^{\frac{3p-2}{2p}} \left(\sum_{i=0}^{k} \frac{A_i}{\lambda_i^{\frac{p}{p-2}}}\right)^{-\frac{p-2}{2p}}. \tag{51}$$

Equation (50) follows from combining (51) with (49).

To end our proof, we use the elementary fact (Bubeck et al., Lem 3.4) that for a positive sequence $B_j$ such that $B_k^\alpha \geq c_2 \sum_{i=1}^{k} B_j$, we have

$$B_k \geq \left(\frac{\alpha-1}{\alpha} c_2 k\right)^{\frac{1}{\alpha-1}}$$

with the identificatons $\alpha = \frac{p}{p-2}$, $B_k = A_k^{\frac{p-2}{3p-2}}$ and $c_2 = \frac{c_1^{-\frac{p-2}{3p-2}}}{4^{\frac{p}{3p-2}}}$. Subsequently,

$$A_k \geq \left(\frac{2c_2 k}{p}\right)^{\frac{3p-2}{2}} = \Theta\left((\delta k)^{\frac{3p-2}{2}} E_0^{-\frac{p-2}{2}}\right),$$

as desired. Picking up the constants, we have the bound

$$f(y_k) - f(x^*) \leq \frac{E_0}{A_k} = \frac{c_3 D_h(x^*, x_0)^{\frac{p}{2}}}{(\delta k)^{\frac{3p-2}{2}}},$$

where $c_3^{-1} = a(2/p)^{\frac{3p-2}{2}} (4/m)^{-\frac{p-2}{2}}$.

### C.4 Restarting Scheme

When $f$ is *strongly smooth* and $\mu$-gradient dominated, we define the restarting scheme (similar to (44)), which proceeds by running Algorithm 2 for some number of iterations at each step,

$$\hat{x}_k = \text{(the output } y_c \text{ of running Algorithm 2 for } c \text{ iterations with input } x_0 = \hat{x}_{k-c}). \qquad (52)$$

We summarize the behavior of the restarting scheme in the following theorem:

**Theorem 15** *Assume $f$ is convex and $s$-strongly smooth of order $1 < p < \infty$ with constants $0 < L_1, \ldots, L_p < \infty$ and $f$ is $\mu$-gradient dominated of order $p$. Take $h(x) = \frac{1}{2}\|x\|^2$. Let $\hat{x}_k$ be the output of running the restarting scheme (52) for $k/c$ times with $c = (p^3/2)^{\frac{p}{3p-2}}(e/3\kappa)^{\frac{2}{3p-2}}$ where $\kappa = \mu\delta^{\frac{3p-2}{2}} = \mu\eta$. Finally, let $y_k$ be the output of running the rescaled gradient descent update one step from $\hat{x}_k$. Then we have the convergence rate upper bound:*

$$f(y_k) - f(x^*) = O\left(\exp\left(-c_1\mu^{\frac{2}{3p-2}}\delta k\right)\right),$$

*where $c_1 = (3/e)^{\frac{2}{3p-2}}(2/p^3)^{\frac{p}{3p-2}}$.*

Take $h(x) = \frac{1}{2}\|x\|^2$ which is 1-strongly convex. Running $k$ iterations of algorithm (46) results in the convergence bound

$$\frac{\mu}{p}\|\hat{x}_k - x^*\|^p \leq f(\hat{x}_k) - f(x^*) \leq \frac{\frac{c_3}{2}\|\hat{x}_{k-c} - x^*\|^p}{(\delta k)^{\frac{3p-2}{2}}} \leq \frac{\frac{c_3}{2}\|\hat{x}_{k-c} - x^*\|^p}{(\delta c)^{\frac{3p-2}{2}}} \leq \frac{\mu}{pe}\|\hat{x}_{k-c} - x^*\|^p, \quad (53)$$

where the last inequality follows from the choice $c = (c_3 pe/2\kappa)^{\frac{2}{3p-2}}$ where $\kappa = \delta^{\frac{3p-2}{2}}\mu$. Thus an execution of (52) for $c$ iterations of the accelerated method reduces the distance to optimum by a factor of at least $1/e$. Iterating (53), we obtain $\frac{1}{p}\|\hat{x}_k - x^*\|^p \leq e^{-k/c}\frac{1}{p}\|\hat{x}_0 - x^*\|^p$. Here, we require that the update from $x_k$ to $y_{k+1}$ be a descent algorithm. Using the descent property for both methods $E_{k+1} \leq \delta 2p^{p-1}\|x_k - x^*\|^p$ (2a) and $E_{k+1} \leq \delta 2p^{p-1}\|x_{k+1} - x^*\|^p$ (2b) implies that

$$f(\hat{y}_k) - f(x^*) \leq \delta 2p^{p-1}e^{-c_4\mu^{\frac{2}{3p-2}}\delta k}\|x_0 - x^*\|^p = O\left(e^{-c_4\mu^{\frac{2}{3p-2}}\delta k}\right),$$

where $c_4 = (c_3 pe/2)^{-\frac{2}{3p-2}}$.

### C.5 Proof of Theorem 10

We show under the strong smoothness, rescaled gradient descent with line search condition (46c) satisfies (46d). We summarize in the following Lemma.

**Lemma 16** *Under the above assumptions, if $\eta^{\frac{1}{p-1}} \leq \min\{\frac{2}{5p}, 1/(2\sum_{m=2}^p \frac{L_m}{m!})\}$ and $\lambda_{k+1}$ is such that*

$$\frac{3}{4} \leq \frac{\lambda_{k+1}\|x_{k+1} - x_k\|^{p-2}}{\eta} \leq \frac{5}{4}, \qquad (54)$$

*then rescaled gradient descent (12) satisfies*

$$\|x_{k+1} - x_k + \lambda_{k+1}\nabla f(x_{k+1})\| \leq \frac{1}{2}\|x_{k+1} - x_k\|. \qquad (55)$$

Note, we can write (54) as

$$\frac{3}{4}\frac{\eta^{\frac{1}{p-1}}}{\|\nabla f(x_k)\|^{\frac{p-2}{p-1}}} \leq \lambda_{k+1} \leq \frac{5}{4}\frac{\eta^{\frac{1}{p-1}}}{\|\nabla f(x_k)\|^{\frac{p-2}{p-1}}}. \qquad (56)$$

Plugging in the RGD update (12) to (55), what we wish to show is that

$$\left\|\lambda_{k+1}\nabla f(x_{k+1}) - \frac{\eta^{\frac{1}{p-1}}}{\|\nabla f(x_k)\|^{\frac{p-2}{p-1}}}\nabla f(x_k)\right\| \leq \frac{\eta^{\frac{1}{p-1}}}{2}\|\nabla f(x_k)\|^{\frac{1}{p-1}}. \qquad (57)$$

Since $\|\nabla^p f(x)\| \leq L_p$, we have the following Taylor expansion of $\nabla f$:

$$\nabla f(x_{k+1}) = \nabla f(x_k) + \sum_{m=2}^{p-1}\frac{1}{(m-1)!}(\nabla^m f(x_k))(x_{k+1} - x_k)^{m-1} + R_k$$

where $R_k$ is the remainder term which can be bounded as

$$\|R_k\| \leq \frac{L_p}{(p-1)!}\|x_{k+1} - x_k\|^{p-1} = \frac{L_p}{(p-1)!}\eta\|\nabla f(x_k)\|.$$

Furthermore, by strong smoothness assumption, for $m = 2, \ldots, p-1$ we have

$$\|(\nabla^m f(x_k))(x_{k+1} - x_k)^{m-1}\| = \eta^{\frac{m}{p-1}} \frac{|(\nabla^m f(x_k))(\nabla f(x_k))^{m-1}|}{\|\nabla f(x_k)\|^{\frac{(m-1)(p-2)}{p-1}}}$$

$$\leq \eta^{\frac{m}{p-1}} \frac{L_m \|\nabla f(x_k)\|^{m-1+\frac{p-m}{p-1}}}{\|\nabla f(x_k)\|^{\frac{(m-1)(p-2)}{p-1}}}$$

$$= \eta^{\frac{m}{p-1}} L_m \|\nabla f(x_k)\|.$$

By plugging in the bounds above to the left-hand side of (57), we get

$$\left\| \lambda_{k+1} \nabla f(x_{k+1}) - \frac{\eta^{\frac{1}{p-1}}}{\|\nabla f(x_k)\|^{\frac{p-2}{p-1}}} \nabla f(x_k) \right\|$$

$$= \left\| \left( \lambda_{k+1} - \frac{\eta^{\frac{1}{p-1}}}{\|\nabla f(x_k)\|^{\frac{p-2}{p-1}}} \right) \nabla f(x_k) + \lambda_{k+1} \sum_{m=2}^{p-1} \frac{1}{(m-1)!} (\nabla^m f(x_k))(x_{k+1} - x_k)^{m-1} + \lambda_{k+1} R_k \right\|$$

$$\leq \left| \lambda_{k+1} - \frac{\eta^{\frac{1}{p-1}}}{\|\nabla f(x_k)\|^{\frac{p-2}{p-1}}} \right| \|\nabla f(x_k)\| + \lambda_{k+1} \sum_{m=2}^{p-1} \frac{1}{(m-1)!} \|(\nabla^m f(x_k))(x_{k+1} - x_k)^{m-1}\| + \lambda_{k+1} \|R_k\|$$

$$\leq \left| \lambda_{k+1} - \frac{\eta^{\frac{1}{p-1}}}{\|\nabla f(x_k)\|^{\frac{p-2}{p-1}}} \right| \|\nabla f(x_k)\| + \lambda_{k+1} \sum_{m=2}^{p-1} \frac{1}{(m-1)!} \eta^{\frac{m}{p-1}} L_m \|\nabla f(x_k)\|_* + \lambda_{k+1} \frac{L_p}{(p-1)!} \eta \|\nabla f(x_k)\|$$

$$= \left( \left| \lambda_{k+1} - \frac{\eta^{\frac{1}{p-1}}}{\|\nabla f(x_k)\|^{\frac{p-2}{p-1}}} \right| + \lambda_{k+1} \sum_{m=2}^{p-1} \frac{\eta^{\frac{m}{p-1}} L_m}{(m-1)!} + \lambda_{k+1} \frac{L_p}{(p-1)!} \eta \right) \|\nabla f(x_k)\|$$

$$= \left( \left| \lambda_{k+1} - \frac{\eta^{\frac{1}{p-1}}}{\|\nabla f(x_k)\|^{\frac{p-2}{p-1}}} \right| + \lambda_{k+1} \sum_{m=2}^{p} \frac{\eta^{\frac{m}{p-1}} m L_m}{m!} \right) \|\nabla f(x_k)\|$$

$$\leq \left( \left| \lambda_{k+1} - \frac{\eta^{\frac{1}{p-1}}}{\|\nabla f(x_k)\|^{\frac{p-2}{p-1}}} \right| + \lambda_{k+1} \eta^{\frac{2}{p-1}} p \sum_{m=2}^{p} \frac{L_m}{m!} \right) \|\nabla f(x_k)\|$$

$$\leq \left( \left| \lambda_{k+1} - \frac{\eta^{\frac{1}{p-1}}}{\|\nabla f(x_k)\|^{\frac{p-2}{p-1}}} \right| + \lambda_{k+1} \frac{\eta^{\frac{1}{p-1}} p}{2} \right) \|\nabla f(x_k)\|$$

where in the last step we have used that $\eta^{\frac{1}{p-1}} \leq 1/(2\sum_{m=2}^{p} \frac{L_m}{m!})$.

Therefore, from the above, we see that if

$$\left| \lambda_{k+1} - \frac{\eta^{\frac{1}{p-1}}}{\|\nabla f(x_k)\|^{\frac{p-2}{p-1}}} \right| \leq \frac{\eta^{\frac{1}{p-1}}}{4\|\nabla f(x_k)\|^{\frac{p-2}{p-1}}} \tag{58}$$

and

$$\lambda_{k+1} \frac{\eta^{\frac{1}{p-1}} p}{2} \leq \frac{\eta^{\frac{1}{p-1}}}{4\|\nabla f(x_k)\|^{\frac{p-2}{p-1}}}, \tag{59}$$

then the desired relation (57) holds. The first condition (58) is equivalent to

$$\frac{3}{4} \frac{\eta^{\frac{1}{p-1}}}{\|\nabla f(x_k)\|^{\frac{p-2}{p-1}}} \leq \lambda_{k+1} \leq \frac{5}{4} \frac{\eta^{\frac{1}{p-1}}}{\|\nabla f(x_k)\|^{\frac{p-2}{p-1}}}$$

which is precisely the requirement (56), whereas the second condition (59) is equivalent to

$$\lambda_{k+1} \leq \frac{1}{2p\|\nabla f(x_k)\|^{\frac{p-2}{p-1}}}.$$

Note that if $\eta^{\frac{1}{p-1}} \leq \frac{2}{5p}$, then the last condition above is automatically satisfied if the right-hand side of the former condition (56) holds. Therefore, we have shown that the condition (56) implies the desired relation (57), or equivalently (55). A simple continuity argument, similar to (Bubeck et al., Lem 3.2) ensures the existence of pair $(\lambda_k, y_k)$ that satisfies (54) and (55) simultaneously.

## C.6 Proximal method

Given $x_k \in \mathbb{R}^n$ and $\eta > 0$, let $x_{k+1}$ be the proximal update (8), which satisfies

$$x_{k+1} = x_k - \eta^{\frac{1}{p-1}} \frac{\nabla f(x_{k+1})}{\|\nabla f(x_{k+1})\|^{\frac{p-2}{p-1}}}. \tag{60}$$

**Lemma 17** *If $\lambda_{k+1}$ is such that*

$$\frac{1}{2} \le \frac{\lambda_{k+1} \|x_{k+1} - x_k\|^{p-2}}{\epsilon} \le \frac{3}{2}, \tag{61}$$

*then*

$$\|x_{k+1} - x_k + \lambda_{k+1} \nabla f(x_{k+1})\| \le \frac{1}{2} \|x_{k+1} - x_k\|. \tag{62}$$

Note (61) is equivalent to the condition

$$\frac{1}{2} \frac{\eta^{\frac{1}{p-1}}}{\|\nabla f(x_{k+1})\|^{\frac{p-2}{p-1}}} \le \lambda_{k+1} \le \frac{3}{2} \frac{\eta^{\frac{1}{p-1}}}{\|\nabla f(x_{k+1})\|^{\frac{p-2}{p-1}}}. \tag{63}$$

Plugging in the proximal update (60) to (62), what we wish to show is that

$$\left\| \lambda_{k+1} \nabla f(x_{k+1}) - \frac{\eta^{\frac{1}{p-1}}}{\|\nabla f(x_{k+1})\|^{\frac{p-2}{p-1}}} \nabla f(x_{k+1}) \right\| \le \frac{\eta^{\frac{1}{p-1}}}{2} \|\nabla f(x_{k+1})\|^{\frac{1}{p-1}}.$$

Equivalently, we wish to show that

$$\left| \lambda_{k+1} - \frac{\eta^{\frac{1}{p-1}}}{\|\nabla f(x_{k+1})\|^{\frac{p-2}{p-1}}} \right| \le \frac{\epsilon^{\frac{1}{p-1}}}{2 \|\nabla f(x_{k+1})\|^{\frac{p-2}{p-1}}},$$

which is exactly condition (63). Subsequently, we can write the Monteiro-Svaiter-style accelerated proximal method as the following sequence of updates,

---

**Algorithm 3** Monteiro-Svaiter-style accelerated proximal method

---

**Require:** $f$ is differentiable and $h$ is 1-strongly convex
1: Set $x_0 = z_0 = 0$, $A_0 = 0$, $\delta^{\frac{3p-2}{2}} = \eta$, $\eta > 0$
2: **for** $k = 1, \ldots, K$ **do**
3: Choose $\lambda_{k+1}$ (e.g. by line search) such that $\frac{1}{2} \le \frac{\lambda_{k+1} \|y_{k+1} - x_k\|^{p-2}}{\eta} \le \frac{3}{2}$, where

$$y_{k+1} = \arg\min_{x \in \mathcal{X}} \left\{ f(x) + \frac{1}{\eta p} \|x - x_k\|^p \right\},$$

and $\alpha_k = \frac{\lambda_{k+1} + \sqrt{\lambda_{k+1} + 4 A_k \lambda_{k+1}}}{2\delta}$, $A_{k+1} = \delta \alpha_k + A_k$, $\tau_k = \frac{\alpha_k}{A_{k+1}}$ (so that $\lambda_{k+1} = \frac{\delta^2 \alpha_k^2}{A_{k+1}}$) and

$$x_k = \delta \tau_k z_k + (1 - \delta \tau_k) y_k.$$

4: Update $z_{k+1} = \arg\min_{z \in \mathcal{X}} \left\{ \alpha_k \langle \nabla f(y_{k+1}), z \rangle + \frac{1}{\delta} D_h(z, z_k) \right\}$
5: **return** $y_K$.

---

# D  Examples and Numerical Experiments

## D.1  Comparison to Runge-Kutta

In Zhang et al. (2018) the following gradient lower bound assumption is made

**Definition 4** *$f$ satisfies the* gradient lower bound *of order $p \ge 2$ if for all $m = 1, \ldots, p - 1$,*

$$f(x) - f(x^*) \ge \frac{1}{C_m} \|\nabla^m f(x)\|^{\frac{p}{p-m}} \quad \forall x \in \mathbb{R}^n$$

*for some constants $0 < C_1, \ldots, C_{p-1} < \infty$.*

Notice that when $p = 2$, this is equivalent to $s$-strong smoothness, which is the general smoothness condition on the gradient. However, for $p > 2$ we can show that it is slightly weaker than strong smoothness. We summarize in the following Lemma:

**Lemma 18** *If $f$ is strongly smooth of order $p$ with constants $L_m$, then $f$ satisfies the gradient lower bound of order $p$ with constants $C_m = 4(\sum_{i=2}^{p} \frac{L_i}{i!})L_m^{\frac{p}{p-m}}$.*

Let $\eta = 1/(2\sum_{m=2}^{p} \frac{L_m}{m!})^{p-1}$ as in (2). Then with $x_k = x$ and $x_{k+1} = x - \eta^{\frac{1}{p-1}}\nabla f(x)/\|\nabla f(x)\|^{\frac{p-2}{p-1}}$, by Lemma 4 we have

$$f(x^*) \leq f(x_{k+1}) \leq f(x) - \frac{\eta^{\frac{1}{p-1}}}{2}\|\nabla f(x)\|^{\frac{p}{p-1}} = f(x) - \frac{1}{4\sum_{m=2}^{p} \frac{L_m}{m!}}\|\nabla f(x)\|^{\frac{p}{p-1}}.$$

Rearranging gives the desired claim:

$$f(x) - f(x^*) \geq \frac{1}{4\sum_{m=2}^{p} \frac{L_m}{m!}}\|\nabla f(x)\|^{\frac{p}{p-1}}.$$

### D.2 Examples

We provide details on the examples presented in the main text.

### D.3 $\ell_p$ loss

Let

$$f(x) = \frac{1}{p}\|x\|_p^p = \frac{1}{p}\sum_{i=1}^{d}|x_i|^p = \frac{1}{p}\sum_{i=1}^{d}\mathrm{sgn}(x_i)^p x_i^p.$$

The gradient $\nabla f(x)$ has entries

$$(\nabla f(x))_i = \mathrm{sgn}(x_i)^p x_i^{p-1}.$$

The norm of the gradient is

$$\|\nabla f(x)\| = \left(\sum_{i=1}^{d} x_i^{2p-2}\right)^{\frac{1}{2}} = \|x\|_{2p-2}^{p-1}.$$

Therefore, for $m \geq 2$,

$$\|\nabla f(x)\|^{\frac{p-m}{p-1}} = \|x\|_{2p-2}^{p-m} = \left(\sum_{i=1}^{d} x_i^{2p-2}\right)^{\frac{p-m}{2p-2}}.$$

For $m \geq 2$, the $m$-th derivative $\nabla^m f(x)$ has nonzero entries only on the diagonal:

$$(\nabla^m f(x))_{i,\dots,i} = (p-1)\cdots(p-m+1)\mathrm{sgn}(x_i)^p x_i^{p-m}.$$

Then for any unit vector $v \in \mathbb{R}^d$,

$$(\nabla^m f(x))(v^m) = (p-1)\cdots(p-m+1)\sum_{i=1}^{d}\mathrm{sgn}(x_i)^p x_i^{p-m} v_i^m.$$

By Hölder's inequality with $q = \frac{2p-2}{p-m}$ and $r = \frac{2p-2}{p+m-2}$, so $\frac{1}{q} + \frac{1}{r} = 1$, we have

$$|(\nabla^m f(x))(v^m)| = (p-1)\cdots(p-m+1)\left|\sum_{i=1}^{d}\mathrm{sgn}(x_i)^p x_i^{p-m} v_i^m\right|$$

$$\leq (p-1)\cdots(p-m+1)\left(\sum_{i=1}^{d}|\mathrm{sgn}(x_i)x_i^{p-m}|^{\frac{2p-2}{p-m}}\right)^{\frac{p-m}{2p-2}}\left(\sum_{i=1}^{d}|v_i^m|^{\frac{2p-2}{p+m-2}}\right)^{\frac{p+m-2}{2p-2}}$$

$$= (p-1)\cdots(p-m+1)\,\|x\|_{2p-2}^{p-m}\left(\sum_{i=1}^{d}|v_i|^{\frac{2m(p-1)}{p+m-2}}\right)^{\frac{p+m-2}{2p-2}}.$$

Note that $\frac{m(p-1)}{p+m-2} = 1 + \frac{(m-1)(p-2)}{p+m-2} \geq 1$. Then using $\sum_{i=1}^{d} c_i^q \leq (\sum_{i=1}^{d} c_i)^q$ for $c_i \geq 0$, $q \geq 1$, we can write

$$\sum_{i=1}^{d}|v_i|^{\frac{2m(p-1)}{p+m-2}} \leq \left(\sum_{i=1}^{d}v_i^2\right)^{\frac{m(p-1)}{p+m-2}} = \|v\|_2^{\frac{2m(p-1)}{p+m-2}} = 1$$

since we assumed $v$ is a unit norm vector, so $\|v\|_2 = 1$. Plugging this to the bound above, we obtain

$$|(\nabla^m f(x))(v^m)| \leq (p-1)\cdots(p-m+1)\,\|x\|_{2p-2}^{p-m}$$
$$= (p-1)\cdots(p-m+1)\,\|\nabla f(x)\|^{\frac{p-m}{p-1}}.$$

Taking the supremum over unit vectors $v \in \mathbb{R}^d$, we conclude that

$$\|\nabla^m f(x)\| \leq (p-1)\cdots(p-m+1)\|\nabla f(x)\|^{\frac{p-m}{p-1}}.$$

This shows that $f$ is strongly smooth of order $p$ with constants

$$L_m = (p-1)\cdots(p-m+1).$$

### D.4 Logistic loss

We show the logistic loss of strongly smooth of order $p = \infty$. We have

$$\nabla f(x) = -\frac{w}{1 + e^{-w^\top x}}$$

and

$$\|\nabla f(x)\| = \frac{\|w\|}{1 + e^{-w^\top x}}.$$

By induction we can see that

$$\nabla^m f(x) = -\frac{(m-1)!w^{\otimes m}}{(1 + e^{-w^\top x})^m}$$

so that

$$\|\nabla^m f(x)\| = \sup_{\|v\|=1}|(\nabla^m f(x))(v^m)| = \frac{(m-1)!\|w\|^m}{(1 + e^{-w^\top x})^m}.$$

Then

$$\frac{\|\nabla^m f(x)\|}{\|\nabla f(x)\|} = \frac{(m-1)!\|w\|^{m-1}}{(1 + e^{-w^\top x})^{m-1}} \leq (m-1)!\|w\|^{m-1}.$$

This shows that $f(x) = \log(1 + e^{-w^\top x})$ satisfies the strong smoothness condition with $p = \infty$ with constant

$$L_m = (m-1)!\|w\|^{m-1}.$$

### D.5 GLM loss

Consider the generalized linear model loss function $f(x) = \frac{1}{2}(y - \phi(x^\top w))^2$ for $\phi(r) = 1/(1 + e^{-r}) \in (0,1)$, $y \in \{0,1\}$, and $w \in \mathbb{R}^d$. Introduce the shorthand $b = 1 - 2y \in \{1,-1\}$, and note that

$$\phi(r) - y = b\phi(br),$$
$$\phi'(r) = e^{-r}/(1 + e^{-r})^2 = \phi(r)\phi(-r) = \phi'(-r) \in (0, 1/4],$$
$$\phi'(r)/\phi(r) = \phi(-r),$$
$$\phi''(r) = \phi'(r)\phi(-r) - \phi(r)\phi'(-r) = \phi'(r)(\phi(-r) - \phi(r)) \in [-1/(6\sqrt{3}), 1/(6\sqrt{3})],$$
$$\phi''(r)/\phi'(r) = \phi(-r) - \phi(r), \quad \text{and}$$
$$\phi'''(r) = \phi''(r)(\phi(-r) - \phi(r)) - 2\phi'(r)^2 \in [-1/2, 0]$$

To simplify the presentation, we will fix $x$ and let $z = x^\top w$. With this notation in place we have

$$f(x) = \tfrac{1}{2}\phi(bz)^2,$$
$$\nabla f(x) = b\phi(bz)\phi'(bz)w,$$
$$\nabla^2 f(x) = (\phi'(bz)^2 + \phi(bz)\phi''(bz))ww^\top, \quad \text{and}$$
$$\nabla^3 f(x) = b(3\phi'(bz)\phi''(bz) + \phi(bz)\phi'''(bz))w^{\otimes 3}.$$

Since $\phi(r)\phi'(r) \in (0,1)$, we have, for any $a \in [0,1]$

$$\frac{\|\nabla^2 f(x)\|}{\|\nabla f(x)\|^a} = \frac{|\phi'(bz)^2 + \phi(bz)\phi''(bz)|}{|\phi(bz)\phi'(bz)|^a}\|w\|^{2-a} \leq \frac{|\phi'(bz)^2 + \phi(bz)\phi''(bz)|}{|\phi(bz)\phi'(bz)|}\|w\|^{2-a}$$
$$= |2\phi(-bz) - \phi(bz)|\|w\|^{2-a} \leq 2\|w\|^{2-a}.$$

Moreover,

$$|\nabla^3 f(x)| = |3\phi'(bz)\phi''(bz) + \phi(bz)\phi'''(bz)|\|w\|^3 \leq (\sqrt{3}/24 + 1/2)\|w\|^3.$$

Therefore, $f$ is s-strongly smooth of order $p = 3$ with $L_2 = 2\|w\|^{1.5}$ and $L_3 = (\sqrt{3}/24 + 1/2)\|w\|^3$.

# E   Additional Results

## E.1   Coordinate Descent Methods

At each iteration, a randomized coordinate method samples a coordinate direction $i \in \{1,\ldots,d\}$ uniformly at random and performs an update along that coordinate direction. Denote $\nabla_{i_k} f = e_{i_k} e_{i_k}^\top \nabla f(x)$ where $e_i$ is the $i$-th basis vector.

**Definition 5** *An algorithm $x_{k+1} = \mathcal{A}(x_k)$ is a* **coordinate descent algorithm of order** $1 < p \leq \infty$, *if for some constant $0 < \delta < \infty$, it almost surely satisfies*

$$\frac{f(x_{k+1}) - f(x_k)}{\delta} \leq -\|\nabla_{i_k} f(x_k)\|_*^{\frac{p}{p-1}}. \tag{65}$$

For coordinate descent methods of order $p$, it is possible to obtain non-asymptotic guarantees for non-convex, convex and gradient dominated functions. We summarize in the following theorems.

**Theorem 19** *Suppose an algorithm satisfies* (65) *for some $0 < \delta < \infty$ and $1 < p \leq \infty$ and $f$ is differentiable. Then the algorithm also satisfies*

$$\min_{0 \leq s \leq k} \mathbb{E}\|\nabla_{i_s} f(x_s)\|_* \leq (E_0/(\delta k))^{\frac{p-1}{p}} = O(1/\delta k). \tag{66}$$

**Theorem 20** *Suppose an algorithm satisfies* (65) *for some $0 < \delta < \infty$ and $1 < p \leq \infty$ and $f$ is differentiable and convex with $R = \sup_{x:f(x)\leq f(x_0)} \|x - x^*\| < \infty$. Then the algorithm satisfies*

$$\mathbb{E}[f(x_k)] - f(x^*) = \begin{cases} O\left(1/\left(1 + \frac{1}{Rp}(\delta k)^{\frac{p-1}{p}}\right)^p\right) & \text{if } p < \infty \\ O\left(e^{-\delta k/R}\right) & \text{if } p = \infty \end{cases}. \tag{67}$$

**Theorem 21** *Suppose an algorithm satisfies* (2) *for some $0 < \delta < \infty$ and $1 < p \leq \infty$, and $f$ is differentiable and $\mu$-gradient dominated of order $p$. Then the algorithm satisfies*

$$\mathbb{E}[f(x_k)] - f(x^*) = O\left(e^{-\frac{1}{d}\frac{p}{p-1}\mu^{\frac{1}{p-1}}\delta k}\right). \tag{68}$$

### E.1.1   Proof of Theorem 19

$$\delta k \mathbb{E} \min_{0 \leq s \leq k} \|\nabla_s f(x_s)\|_*^{\frac{p}{p-1}} \leq \mathbb{E} \sum_{s=0}^{k} \|\nabla_s f(x_s)\|_*^{\frac{p}{p-1}}\delta \leq f(x_0) - \mathbb{E}f(x_k) \leq f(x_0)$$

Rearranging the inequality yields the result in Theorem 19.

### E.1.2 Proof of Theorem 20

For the proof of Theorem 20 under the condition (65), we use the energy function
$$E_k = w_a(\delta k)(f(x_k) - f(x^*)),$$
When (65) holds, we have

$$
\begin{aligned}
\frac{E_{k+1}-E_k}{\delta} &= \frac{w_a(\delta(k+1))-w_a(\delta k)}{\delta}(f(x_k)-f(x^*)) + w_a(\delta(k+1))\frac{f(x_{k+1})-f(x_k)}{\delta}\\
&\leq \frac{w_a(\delta(k+1))-w_a(\delta k)}{\delta}\langle\nabla f(x_k), x_k - x^*\rangle + w_a(\delta(k+1))\frac{f(x_{k+1})-f(x_k)}{\delta}\\
&\overset{(65)}{\leq} \frac{w_a(\delta(k+1))-w_a(\delta k)}{\delta}\langle\nabla f(x_k), x_k - x^*\rangle - w_a(\delta(k+1))\|\nabla_{i_k}f(x_k)\|_*^{\frac{p}{p-1}}\\
&= w_a(\delta(k+1))\left(\frac{w_a(\delta(k+1))-w_a(\delta k)}{\delta w_a(\delta(k+1))}\langle\nabla f(x_k), x_k - x^*\rangle - \|\nabla_{i_k}f(x_k)\|_*^{\frac{p}{p-1}}\right)\\
&\leq w_a(\delta(k+1))\left(\frac{1}{aw_a(\delta(k+1))^{1/p}}\langle\nabla f(x_k), x_k - x^*\rangle - \|\nabla_{i_k}f(x_k)\|_*^{\frac{p}{p-1}}\right)\\
&= w_a(\delta(k+1))\left(\frac{1}{aw_a(\delta(k+1))^{1/p}}\langle\nabla_{i_k}f(x_k), x_k - x^*\rangle - \|\nabla_{i_k}f(x_k)\|_*^{\frac{p}{p-1}}\right) + \xi_k\\
&\leq w_a(\delta(k+1))c_p\|\frac{1}{aw_a(\delta(k+1))^{1/p}}(x_k - x^*)\|^p + \xi_k\\
&= c_p\|x_k - x^*\|^p/a^p + \xi_k \leq c_p R^p/a^p + \xi_k.
\end{aligned}
$$

Here, the martingale $\xi_k := \frac{w_a(\delta(k+1))}{aw_a(\delta(k+1))^{1/p}}\langle\nabla f(x_k) - \nabla_{i_k}f(x_k), x_k - x^*\rangle$. The first inequality uses convexity of $f$, and the second uses (2a). The third inequality is an application of (33). The fourth inequality uses the Fenchel-Young inequality with $s = \nabla_{i_k}f(x_k)$ and $u = \frac{1}{aw_a(\delta(k+1))^{1/p}}(x_k - x^*)$. Both descent conditions (2) imply $\|x_k - x^*\| \leq R$, yielding the final inequality. Therefore, we have shown that for all $k \geq 0$, $\mathbb{E}[E_{k+1}|x_k] - E_k \leq c_p\delta R^p/a^p$. This implies $\mathbb{E}[E_k] \leq E_0 + c_p\delta k R^p/a^p$. Therefore

$$\mathbb{E}[f(x_k)] - f(x^*) \leq \frac{f(x_0)-f(x^*)}{(1+\delta k/(ap))^p} + c_p\frac{R^p}{a^p}\frac{\delta k}{(1+\delta k/(ap))^p}.$$

Since $a > 0$ was arbitrary, we may choose $a = R\frac{(c_p\delta k)^{1/p}}{(f(x_0)-f(x^*))^{1/p}}$ to obtain the bound

$$\mathbb{E}[f(x_k)] - f(x^*) \leq \frac{2(f(x_0)-f(x^*))}{\left(1+\frac{(f(x_0)-f(x^*))^{1/p}}{Rc_p^{1/p}p}(\delta k)^{\frac{p-1}{p}}\right)^p} = O(1/(1 + \frac{1}{Rp}(\delta k)^{\frac{p-1}{p}})^p)$$

as desired.

### E.1.3 Proof of Theorem 21

Take the energy function $E_k = f(x_k) - f(x^*)$, and observe that if (2a) holds, then we have:

$$
\begin{aligned}
\frac{\mathbb{E}[E_{k+1}|x_k]-E_k}{\delta} = \frac{\mathbb{E}[f(x_{k+1})|x_k]-f(x_k)}{\delta} &\overset{(65)}{\leq} -\mathbb{E}[\|\nabla_{i_k}f(x_k)\|_*^{\frac{p}{p-1}}|x_k]\\
&= -\frac{1}{d}\sum_{i=1}^d\|\nabla_i f(x_k)\|^{\frac{p}{p-1}}\\
&\leq -\frac{1}{d}\|\nabla f(x_k)\|^{\frac{p}{p-1}}\\
&\overset{(3)}{\leq} -\frac{1}{d}\frac{p}{p-1}\mu^{\frac{1}{p-1}}E_k,
\end{aligned}
$$

or rewritten, $\mathbb{E}[E_{k+1}] \leq \left(1 - \frac{1}{d}\frac{p}{p-1}\mu^{\frac{1}{p-1}}\delta\right)E_k$. Summing gives the bound

$$\mathbb{E}[E_{k+1}] \leq \left(1 - \frac{1}{d}\frac{p}{p-1}\mu^{\frac{1}{p-1}}\delta\right)^k E_0 \leq e^{-\frac{1}{d}\frac{p}{p-1}\mu^{\frac{1}{p-1}}\delta k}E_0,$$

### E.1.4 Rescaled coordinate descent

Rescaled coordinate descent,

$$x_{k+1} = x_k - \eta_{i_k}^{\frac{1}{p-1}}\frac{\nabla_{i_k}f(x_k)}{\|\nabla_{i_k}f(x_k)\|^{\frac{p-2}{p-1}}} = \arg\min_{x\in\mathcal{X}}\left\{\langle\nabla_{i_k}f(x_k), x\rangle + \frac{1}{\eta_{i_k}p}\|x - x_k\|^p\right\} \qquad (69)$$

where $0 < \eta_{i_k} < \infty$ for $i_k \in \{1,\dots k\}$, satisfies (65) provided the objective is strongly smooth along each coordinate direction.

**Definition 6** *A function $f$ is **strongly smooth** of order $p$ along each coordinate direction for $p > 1$, if there exist constants $0 < L_1^{(i)}, \ldots, L_p^{(i)} < \infty$ for $i = 1, \ldots, d$, such that for $m = 1, \ldots, p-1$ and for all $x \in \mathbb{R}^d$, as well as for all $i \in \{1, \ldots d\}$*

$$\nabla^m f(x)(\nabla_i f(x))^m \leq L_m^{(i)} \|\nabla_i f(x)\|_*^{m + \frac{p-m}{p-1}}, \tag{70}$$

*and moreover for $m = p$, $f$ satisfies the condition $\|\nabla^p f(x)\| \leq L_p^{(i)}$.*

We summarize our results regarding the rescaled coordinate descent in the following Lemma.

**Lemma 22** *Suppose $f$ is strongly smooth of order $p \geq 2$ along each coordinate direction with constants $0 < L_1^{(i)}, \ldots, L_p^{(i)} < \infty$ for $i = 1, \ldots, d$. Then rescaled gradient descent (69) with step size*

$$0 < \eta_i^{\frac{1}{p-1}} \leq \min\left\{1, \frac{1}{\left(2 \sum_{m=2}^p \frac{L_m^{(i)}}{m!}\right)}\right\} \tag{71}$$

*satisfies (65) with $\delta = \min_{i=1,\ldots,d} \eta_i^{\frac{1}{p-1}}/2$.*

### E.2 Accelerating Coordinate Descent Methods

Coordinate descent algorithms of order $p$ can also be accelerated. Suppose $f$ is convex. Set $A_k = C\delta^p k^{(p)}$ where we use the rising factorial $k^{(p)} = k(k+1)\cdots(k+p-1)$. Denote $\alpha_k := \frac{A_{k+1} - A_k}{\delta} = Cp\delta^{p-1}(k+1)^{(p-1)}$ and $\tau_k := \frac{\alpha_k}{A_{k+1}} = \frac{k}{\delta(k+p)}$. We write the algorithm as,

$$x_k = \delta\tau_k z_k + (1 - \delta\tau_k)y_k \tag{72a}$$

$$z_{k+1} = \arg\min_z \left\{\alpha_k \langle \nabla_{i_k} f(x_k), z \rangle + \frac{1}{\delta} D_h(z, z_k)\right\} \tag{72b}$$

where the update for $y_{k+1}$ satisfies the descent condition

$$\frac{f(y_{k+1}) - f(x_k)}{\delta^{\frac{p}{p-1}}} \leq -\|\nabla_{i_k} f(x_k)\|^{\frac{p}{p-1}}. \tag{73}$$

For algorithm (72), using (38) we compute

$$\frac{E_{k+1} - E_k}{\delta} = \frac{D_h(x^*, z_{k+1}) - D_h(x^*, z_k)}{\delta} + \frac{A_{k+1}}{\delta}(f(y_{k+1}) - f(x^*)) - \frac{A_k}{\delta}(f(y_k) - f(x^*)). \tag{74}$$

We bound the first part,

$$
\begin{aligned}
\frac{D_h(x^*, z_{k+1}) - D_h(x^*, z_k)}{\delta} &= -\left\langle \frac{\nabla h(z_{k+1}) - \nabla h(z_k)}{\delta}, x^* - z_{k+1} \right\rangle - \frac{1}{\delta} D_h(z_{k+1}, z_k) \\
&\stackrel{(72b)}{=} \alpha_k \langle \nabla_{i_k} f(x_k), x^* - z_k \rangle + \alpha_k \langle \nabla_{i_k} f(x_k), z_k - z_{k+1} \rangle \\
&\quad - \frac{1}{\delta} D_h(z_{k+1}, z_k) \\
&\leq \alpha_k \langle \nabla f(x_k), x^* - z_k \rangle - \xi_k - (\delta/m)^{\frac{1}{p-1}} \alpha_k^{\frac{p}{p-1}} \|\nabla_i f(x_k)\|^{\frac{p}{p-1}}, \tag{75}
\end{aligned}
$$

where $\xi_k = \alpha_k \langle \nabla f(x_k) - \nabla_{i_k} f(x_k), x^* - z_k \rangle$ which is a martingale. The inequality follows from the $m$-uniform convexity of $h$ of order $p$ and the Fenchel-Young inequality $\langle s, u \rangle + \frac{1}{p}\|u\|^p \geq -\frac{p}{p-1}\|s\|_*^{\frac{p}{p-1}}$, with $u = (m/\delta)^{\frac{1}{p}}(z_{k+1} - z_k)$ and $s = (\delta/m)^{\frac{1}{p}} \alpha_k^{\frac{p}{p-1}} \nabla_{i_k} f(x_k)$. Plugging in update (15a),

$$
\begin{aligned}
\alpha_k \langle \nabla f(x_k), x^* - z_k \rangle &= \alpha_k \langle \nabla f(x_k), x^* - y_k \rangle + \frac{A_{k+1}}{\delta} \langle \nabla f(x_k), y_k - x_k \rangle \\
&= \alpha_k \langle \nabla f(x_k), x^* - x_k \rangle + \frac{A_k}{\delta} \langle \nabla f(x_k), y_k - x_k \rangle \\
&\leq -\left(\frac{A_{k+1}}{\delta}(f(y_{k+1}) - f(x^*)) - \frac{A_k}{\delta}(f(y_k) - f(x^*))\right) \\
&\quad + A_{k+1} \frac{f(y_{k+1}) - f(x_k)}{\delta} \\
&\stackrel{(73)}{\leq} -\left(\frac{A_{k+1}}{\delta}(f(y_{k+1}) - f(x^*)) - \frac{A_k}{\delta}(f(y_k) - f(x^*))\right)
\end{aligned}
$$

$$- A_{k+1} \delta^{\frac{1}{p-1}} \| \nabla_{i_k} f(x_k) \|^{\frac{p}{p-1}}. \tag{76}$$

The first inequality follows from the convexity of $f$ and rearranging terms. The second inequality uses (73). Combining (74) with (75) and (76) we have,

$$\frac{E_{k+1} - E_k}{\delta} \leq \left( (\delta/m)^{\frac{1}{p-1}} (Cp\delta^{p-1}(k+1)^{(p-1)})^{\frac{p}{p-1}} - C\delta^{\frac{1}{p-1}} \delta^p (k+1)^{(p)} \right) \| \nabla_{i_k} f(x_k) \|^{\frac{p}{p-1}} - \xi_k.$$

Given $((k+1)^{(p-1)})^{\frac{p}{p-1}}/(k+1)^{(p)} \leq 1$, it suffices that $C \leq 1/mp^p$ to ensure $\frac{\mathbb{E}[E_{k+1}|x_k] - E_k}{\delta} \leq 0$. Summing, we obtain the desired bound.

$$\mathbb{E}[f(x_k)] - f(x^*) \lesssim 1/(\delta k)^p.$$

### E.2.1 Accelerating rescaled coordinate descent

A corollary to the coordinate descent property of rescaled descent with step size (71) is that it can be combined with sequences (72a) and (72b) to form a method with an $O(1/(\delta k)^p)$ convergence rate upper bound. We summarize this result in the following theorem.

---

**Algorithm 4** Nesterov-style accelerated rescaled coordinate descent.

---

**Require:** $f$ is *strongly smooth of order $p$* along each coordinate direction and $h$ satisfies $D_h(x, y) \geq \frac{1}{p}\|x - y\|^p$.

1: Set $x_0 = z_0 = 0$ and $A_k = C\delta^p k^{(p)}$, $\alpha_k = \frac{A_{k+1} - A_k}{\delta} = Cp\delta^{p-1}(k+1)^{(p-1)}$ and $\tau_k = \frac{\alpha_k}{A_{k+1}} = \frac{k}{\delta(k+p)}$ where $k^{(p)} := k(k+1)\cdots(k+p-1)$.

2: **for** $k = 1, \ldots, K$ **do**

3: $\quad x_k = \delta\tau_k z_k + (1 - \delta\tau_k)y_k$

4: $\quad$ sample $i_k \in \{1, \ldots, d\}$. Update

5: $\quad z_{k+1} = \arg\min_z \left\{ \alpha_k \langle \nabla_{i_k} f(x_k), z \rangle + \frac{1}{\delta} D_h(z, z_k) \right\}$

6: $\quad y_{k+1} = x_k - \eta_{i_k}^{\frac{1}{p-1}} \frac{\nabla_{i_k} f(x_k)}{\|\nabla_{i_k} f(x_k)\|_*^{\frac{p-2}{p-1}}}$

7: **return** $y_K$.

---

**Theorem 23** *Suppose $f$ is convex and strongly smooth of order $1 < p < \infty$ along each coordinate direction $i$ with constants $0 < L_1^{(i)}, \ldots, L_p^{(i)} < \infty$. Also suppose $\eta_i$ satisfies (71). Then Algorithm 4 satisfies,*

$$\mathbb{E}[f(y_k)] - f(x^*) \lesssim 1/(\delta k)^p.$$

### E.3 Optimal Universal Higher-order Tensor Methods

We say that it has Hölder continuous $(p-1)$-st order gradients of degree $\nu \in [0, 1]$ on a convex set $\mathcal{X} \subseteq \text{dom} f$, if for some constant $L_\nu$ it holds

$$\|\nabla^{p-1} f(x) - \nabla^{p-1} f(y)\| \leq L_\nu \|x - y\|^\nu \tag{77}$$

The final result of our paper contains the analysis of the following optimal algorithm for minimizing functions that satsify (77)

---

**Algorithm 5** Monteiro-Svaiter-style universal higher-order tensor method.

---

**Require:** $f$ satisfies (77) with parameters $p$ and $L_\nu$, $h$ is 1-strongly convex, $B = I$, $\tilde{p} = p - 1 + \nu$.

1: Set $x_0 = z_0 = 0$, $A_0 = 0$, $\delta^{\frac{3p-2}{2}} = \eta$, $\eta = L_\nu/(p-2)!$
2: **for** $k = 1, \ldots, K$ **do**
3: Choose $\lambda_{k+1}$ (e.g. by line search) such that

$$\frac{1}{2} \leq \frac{\lambda_{k+1}\|y_{k+1}-x_k\|^{\tilde{p}-2}}{\eta} \leq \frac{3}{4}, \tag{78a}$$

where

$$y_{k+1} = \arg\min_{x \in \mathcal{X}} \left\{ f_{p-1}(x; x_k) + \frac{1}{\tilde{p}\eta}\|x - x_k\|^{\tilde{p}} \right\}, \tag{78b}$$

and $\alpha_k = \frac{\lambda_{k+1}+\sqrt{\lambda_{k+1}+4A_k\lambda_{k+1}}}{2\delta}$, $A_{k+1} = \delta\alpha_k + A_k$, $\tau_k = \frac{\alpha_k}{A_{k+1}}$ (so that $\lambda_{k+1} = \frac{\delta^2\alpha_k^2}{A_{k+1}}$) and

$$x_k = \delta\tau_k z_k + (1 - \delta\tau_k)y_k.$$

4: Update $z_{k+1} = \arg\min_{z \in \mathcal{X}} \left\{ \alpha_k \langle \nabla f(y_{k+1}), z \rangle + \frac{1}{\delta}D_h(z, z_k) \right\}$
5: **return** $y_K$.

---

We summarize results on performance of Algorithm 5 in the following corollary to Theorem 9:

**Theorem 24** *Assume $f$ is convex and has Hölder continuous $(p-1)$-st order gradients. Then Algorithm 5 satisfies the convergence rate upper bound*

$$f(y_k) - f(x^*) = O\left(1/(\delta k)^{\frac{3(p-1+\nu)-2}{2}}\right).$$

To prove Theorem 24, the first thing to notice is that the proof of Theorem 9 holds for all $\mathbb{R} \ni p > 0$. Subsequently, to extend our analysis to Algorithm (5), it is sufficient to show (1) (78b) with the line search step (78a) satisfies

$$\|y_{k+1} - x_k - \lambda_{k+1}\nabla f(y_{k+1})\| \leq \frac{1}{2}\|y_{k+1} - x_k\| \tag{79}$$

and that (2) there exists a sequence $(\lambda_{k+1}, y_{k+1})$ that satisfies (78b) and (78a) simultaneously.

**(1)** Observe that the optimality condition for (78b) satisfies

$$\nabla f_{p-1}(y_{k+1}; x_k) - \frac{1}{\eta}(y_{k+1} - x_k)\|y_{k+1} - x_k\|^{\tilde{p}-2} = 0.$$

so that $\|\nabla f_{p-1}(y_{k+1}; x_k)\| = \frac{1}{\eta}\|y_{k+1} - x_k\|^{\tilde{p}-1}$. In particular,

$$y_{k+1} - x_k + \lambda_{k+1}\nabla f(y_{k+1}) = \lambda_{k+1}\nabla f(y_{k+1}) - \frac{\eta}{\|y_{k+1} - x_k\|^{\tilde{p}-2}}\nabla f_{p-1}(y_{k+1}; x_k).$$

From the integral form of the mean value theorem it follows that

$$\|\nabla f_{p-1}(y; x) - \nabla f(y)\| \leq \frac{L_\nu}{(p-2)!}\|y - x\|^{p-2+\nu}.$$

Subsequently

$$\|y_{k+1} - x_k + \lambda_{k+1}\nabla f(y_{k+1})\| \leq \lambda_{k+1}\frac{L_\nu}{(p-2)!}\|y_{k+1} - x_k\|^{\tilde{p}-1} + \left|\lambda_{k+1} - \frac{\eta}{\|y_{k+1}-x_k\|^{\tilde{p}-2}}\right|\|\nabla f_{p-1}(y_{k+1}; x_k)\|$$

$$\leq \|y_{k+1} - x_k\|\left(\lambda_{k+1}\frac{L_\nu}{(p-2)!}\|y_{k+1} - x_k\|^{\tilde{p}-2} + |\frac{\lambda_{k+1}}{\eta}\|y_{k+1} - x_k\|^{\tilde{p}-2} + 1|\right)$$

If we choose $\eta = L_\nu/(p-2)!$ and plug in our line search criterion (78a), we see condition (79) is met.

**(2)**   We now show there exists a pair $(\lambda_{k+1}, y_{k+1})$ that satisfies (78b) and (78a) simultaneously. This claim follows directly form the argument given by Bubeck et al (Bubeck et al., Sec 3.2), which did not rely on $p > 0$ being an integer. For self-containment, we reproduce the argument here.

**Lemma 25** *Let $A \geq 0$, $x, y \in \mathbb{R}^d$ such that $f(x) \neq f(x^*)$. Define the following functions:*

$$a(\lambda) = \frac{\lambda + \sqrt{\lambda^2 + 4\lambda A}}{2}$$

$$x(\lambda) = \frac{a(\lambda)}{A + a(\lambda)} x + \frac{A}{A + a(\lambda)} y$$

$$y(z) = \arg\min_{x \in \mathcal{X}} \left\{ f_{p-1}(w; z) + \frac{1}{\tilde{p}\eta} \|w - z\|^{\tilde{p}} \right\}$$

$$g(\lambda) = \lambda \|y(x(\lambda)) - x(\lambda)\|^{\tilde{p}-1}.$$

*Then we have $g(\mathbb{R}_+) = \mathbb{R}_+$.*

The first claim is that $g(\lambda)$ is a continuous function of $\lambda$. This follows from the fact that $y(z)$ is a continuous function of $z$. Furthermore, $g(0) = 0$, and since $f(x) \neq f(x^*)$ we also have $y(x) \neq x$ which proves $g(+\infty) = +\infty$

**Remark 3** *The same binary line search step introduced by Bubeck et al., Sec 4 finds a $\lambda_{k+1}$ satisfying (78a). The argument given there did not rely on the fact that $p \in \mathbb{Z}_+$.*