[Reviews · NeurIPS 2019]

Reviewer 1



I have read the authors’ response. All the problems have been addressed. However, the quality of the current version cannot meet NeurIPS, because many typos and confusions are arisen, more importantly, the experiments is too simple and it does not enough to support the advantages of the proposed methods. If the paper can be improved accordingly in the final version, it still can be considered to accept.

Reviewer 2



ORIGINALITY. I think the first part of the paper has very good original contributions with correct and nicely-written proofs in the appendix. However, I have the following questions regarding the parts of the paper starting at Section 3. Sorry if these are redundant questions with obvious answers that I missed. 1. The RGD framework is mentioned for both convex and non-convex functions (Lemma 4 doesn't require f to be convex). However, the examples provided are all convex functions, and the focus also seems to be quite heavily on convex functions (because none of the papers on nonconvex optimization are compared with). Do the authors have (1) theoretical results and comparisons with existing work and/or (2)experiments, for non-convex functions? 2. I am unable to appreciate the novelty of the acceleration for RGD since, as mentioned by the authors, previous works by Allen-Zhu/Orecchia, Lessard/Recht/Packard, Lin/Mairal/Harchaoui, and Wilson/Recht/Jordan already provide generalized acceleration for convex functions. Could the authors please make it a little more clear how the acceleration in this work differs from those in these works? QUALITY. I think it's a very high-quality paper. I checked some of the proofs in the appendix, and they are correct. The write-up is also very polished. There was only one place I found a LaTeX error (page 2 of the supplement, when citing Fenchel-Young equality). CLARITY. The paper and proofs are very clearly written! No complaints at all. However, I have one request, which is to (if possible) provide some intuition for the choice of energy function used in the proofs of Theorems 1 - 3. SIGNIFICANCE. See "ORIGINALITY". I don't completely follow the novelty (and, therefore, significance) of the acceleration methods provided, but do think that the first part of the paper (until Section 3) with the RGD framework is significant in unifying many common first-order methods.

Reviewer 3



The paper is generally well written and the mathematics appears to be sound. The content is likely to be of interest to the community.

[Author Response · NeurIPS 2019]

# 1 Summary

We appreciate the reviewers feedback! Generally, the reviewers suggestions could be decomposed into three categories: adding a related works section, cleaning up some of the notation, and clarifying and improving some of the examples and experiments. We address all three of these concerns below.

## 1.1 Adding a related works section

**Response**   To address the concerns of reviewers 1 and 2, we will add a related work section to discuss how our framework relates to other acceleration frameworks proposed, including the ones listed by reviewer 2, and to clarify our contributions. In short, our work can be viewed as a kind of generalization of Allen-Zhu/Orecchia, Lessard/Recht/Packard, Lin/Mairal/Harchaoui, to more general $p > 2$, which allow us to obtain methods with faster convergence guarantees. Our work most closely resembles Wilson/Recht/Jordan, however we (1) introduce and discuss descent methods (which is omitted by Wilson/Recht/Jordan), and (2) provide a general description and Lyapunov analysis of the Monteiro-Svaiter acceleration framework. These manifold generalizations are what allow us to propose a novel method for optimization – RGD and ARGD – which has superior theoretical and empirical performance to several existing methods. We hope to make this clear in the related work section.

## 1.2 Clearing up notation

**Response**   Reviewer 1: Equations (9) and (12) do indeed "hold" – perhaps the confusion is that we did not define $\| \cdot \|_{x_k}$, which we will add. We set the $B$ equal to the identity for the final three examples because for these examples, there is a natural definition of a norm given by the Hessian of the matrix $h$. We will change Lemma 4 to theorem 4. We will also add more details to the proofs so that it is easier to follow (although it would be helpful to understand which proofs the reviewer felt should be more detailed). Reviewer 3: We will add all notational suggestions. Thank you for the helpful clarifying suggestions.

## 1.3 Improving Examples and Experiments

**Response**   Reviewer 1: We will clarify the notation DD and add a more detailed description of experimental results. Reviewer 2: The GLM loss (example 8) is a non-convex function. We are currently running experiments on MNIST for this objective and will add the results to the final version of our paper. As a preview, Hazan et al [Hazan et al., 2015, fig 2] showed that the version of RGD (a.k.a stochastic normalized gradient descent) outperformed stochastic AGD on this objective. We hope to replicate this result in the deterministic setting to confirm our theoretical findings. Reviewer 3: You make an excellent point about the axis. Since we ran the experiment for a fixed $10^{-6}$ iterations, we simply plotted the results without paying attention to whether it dropped below machine precision. We will cut off the plot at the stated $10^{-20}$.

# References

Elad Hazan, Kfir Y. Levy, and Shai Shalev-Shwartz. Beyond convexity: Stochastic quasi-convex optimization. In *Advances in Neural Information Processing Systems 28: Annual Conference on Neural Information Processing Systems 2015, December 7-12, 2015, Montreal, Quebec, Canada*, pages 1594–1602, 2015.


[Meta-Review · NeurIPS 2019]

Dear Authors, Your paper, along with the response, has addressed all reviewers concerns, and this paper is considered as a proposal for acceptance Best